# ADAPTIVE LENGTH IMAGE TOKENIZATION VIA RECURRENT ALLOCATION

**Shivam Duggal    Phillip Isola    Antonio Torralba    William T. Freeman**
MIT CSAIL

## ABSTRACT

Current vision systems typically assign fixed-length representations to images, regardless of the information content. This contrasts with human intelligence —and even large language models—which allocate varying representational capacities based on entropy, context and familiarity. Inspired by this, we propose an approach to learn variable-length token representations for 2D images. Our encoder-decoder architecture recursively processes 2D image tokens, distilling them into 1D latent tokens over multiple iterations of recurrent rollouts. Each iteration refines the 2D tokens, updates the existing 1D latent tokens, and adaptively increases representational capacity by adding new tokens. This enables compression of images into a variable number of tokens, ranging from 32 to 256. We validate our tokenizer using reconstruction loss and FID metrics, demonstrating that token count aligns with image entropy, familiarity and downstream task requirements. Recurrent token processing with increasing representational capacity in each iteration shows signs of token specialization, revealing potential for object / part discovery. Code available at https://github.com/ShivamDuggal4/adaptive-length-tokenizer.

## 1   INTRODUCTION

Representation learning (Bengio et al., 2013), which involves extracting meaningful and useful information from input observations, is crucial for decision-making. An effective representation should be compact while encoding all relevant information. However, what constitutes "relevant" information varies based on the specific task; for example, a coarse classification task may require a different latent representation compression factor for satisfactory performance compared to a task demanding perfect pixel-level reconstruction, which necessitates denser representations. This notion of a useful representation aligns closely with aspects of human intelligence (Legg & Hutter, 2007), particularly the concept of adaptive and variable-compressible representations (Hutter, 2006). Similarly, language models can describe content at various levels of abstraction depending on complexity, context (Graves, 2016; Dehghani et al., 2018), and familiarity (Baevski & Auli, 2018). In contrast, most current visual systems, such as VAEs, VQGANs, and ViTs (Kingma & Welling, 2022; Esser et al., 2020; Dosovitskiy et al., 2020), generate fixed-size representations for all images. In this work, we take a step toward learning adaptive and variable-length visual representations, emphasizing that each image requires a different representation capacity (see Sec. 4).

A common framework for learning image embeddings or representations is the encoder-decoder approach, where an encoder compresses input data into a compact latent representation, which can later be decoded and compared with the original image as a learning objective. While there are other encoder-only methods, such as contrastive learning (Chen et al., 2021) and self-distillation (Caron et al., 2021), we focus on encoder-decoder approaches because a reconstruction objective intuitively promotes the learning of adaptive representations by capturing varying level-of-details necessary for better reconstruction. The current state-of-the-art (transformer-based) encoder-decoder approaches (Dosovitskiy et al., 2020) operate in the discrete token space, by encoding images into learned tokens and then decoding them back to image pixels. To generate these tokens, these approaches compress (slightly) at the input patch-level and then maintain the number of tokens (= number of patches) throughout the encoder-decoder network depth. Thus, the representation length for all images is fixed to the number of tokens, equivalent to the fixed patch-size decided by the human-engineer. Moreover, by having number of tokens equal to number of patches, such approaches are tied to the natural 2D inductive bias of images, preventing any form of adaptive representation or compression

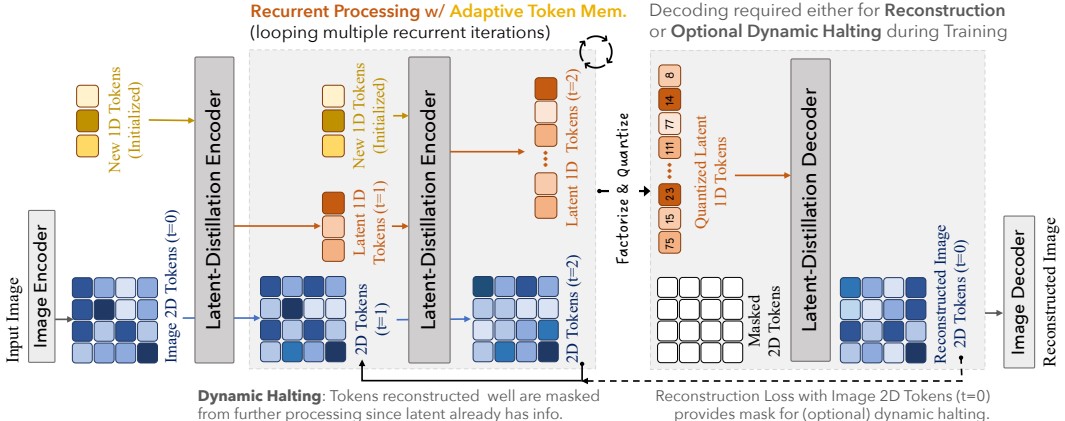

Figure 1: **Adaptive Length Image Tokenizer (ALIT):** Given an image, we first convert it into 2D image tokens before applying the 2D → 1D latent distillation. ALIT recurrently distills 2D image tokens into variable 1D latent tokens, with each iteration adding new latent tokens and processing them with the existing 2D image tokens and the old latent tokens. Training focuses on reconstructing 2D image tokens through reverse distillation from latent 1D to masked 2D tokens. Based on token-reconstruction quality, we can optionally mask specific 2D tokens from further processing, enabling dynamic halting per token. **Recurrent processing with Adaptive Memory** leads to compressible representations, flexible tokenization & specialized tokens focusing on objects/parts.

of different images. Moving away from this inductive bias and with the goal of having modality-agnostic architecture, Google DeepMind proposed Perceiver (Jaegle et al., 2021b;a), a transformer-based architecture which distills input data tokens to a set of fixed 1D tokens. This process of **latent-token distillation** refers to compressing a higher-dimensional input (e.g., 2D image tokens) into a more compact set of latent variables (1D tokens), capturing the most relevant features. Like Perceiver, we also fall into the category of latent-token distillation, where we encode 2D image tokens into much fewer 1D latent tokens via a self-supervised reconstruction objective. While 1D-tokenization of an image overcomes the patch to token constraint and allows much more efficient compression of the input image, a more universal tokenizer would be one which adaptively assigns variable tokens to each input based on content entropy, familiarity etc (Sec. 4).

We tackle the challenge of adaptive or variable-length representation learning by auto-regressively distilling input visual observations into an increasing number of 1D latent tokens. To achieve this, we draw inspiration from foundational works on recurrent computation (Graves, 2016; Dehghani et al., 2018). Recurrent neural networks are often viewed as adaptive thinking modules (Schwarzschild et al., 2021), capable of enhancing the computational requirements of a specific input through recursive processing with the same neural network architecture. Thus, unlike the Matryoshka style (Kusupati et al., 2022) approach of learning multiple representations of varying lengths simultaneously in one-go, we adopt a recurrent computing approach for visual representation learning. In our framework, recurrent computing involves recursively distilling an input image or 2D image tokens into 1D latent tokens through a shared encoder-decoder architecture until each image token has been sufficiently processed/distilled into the latent tokens. At each iteration of this recurrent rollout, *we provide additional computational resources in the form of new learnable latent tokens, enabling the model to learn adaptive and variable-length representations across different iterations.*

We refer to our approach as **ALIT** (**A**daptive **L**ength **I**mage **T**okenizer), and train it using self-supervised image reconstruction objective. Credited to the increasing representational capacity, *each recurrent update leads to the latent tokens specializing and attending to localized regions, hinting at object / part discovery* (see, Fig. 7, Fig. 8, Appendix Fig. 15 and Fig. 16). We validate the effectiveness of the learned tokenizer by demonstrating comparable reconstruction metrics (L1 loss and FID) on multiple datasets (IN, COCO, Places, Art-dataset and even randomly selected internet images Fig. 20) and linear probing results on ImageNet-1K, relative to the 2D VQGAN tokenizer (Esser et al., 2020) and fixed-latent 1D tokenizer, Titok (Yu et al., 2024), while also allowing for flexible token counts per image. By utilizing variable representations per image, we introduce cumulative dataset representations & emphasize the key aspects of effective representations: *required capacity aligns w/ image's information entropy, familiarity, & knowledge of downstream tasks / models.*

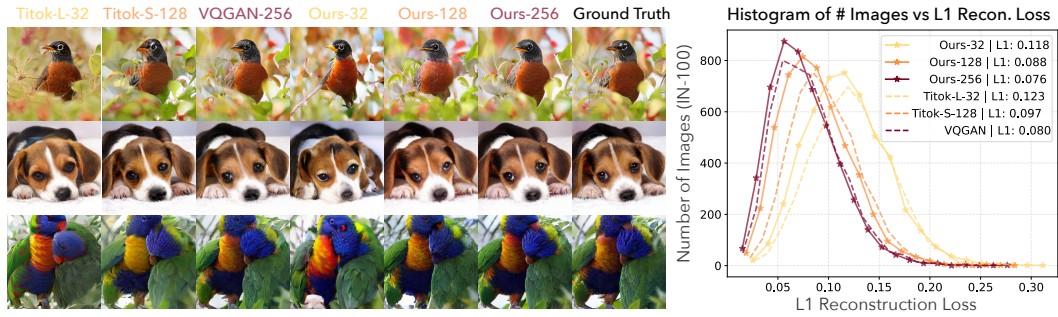

Figure 2: **Reconstruction Analysis on ImageNet-100:** Our approach outperforms all baselines in terms of reconstruction loss (right). Comparing Row-1 (high complexity) and Row-2 (low complexity) demonstrates the effectiveness of adaptive tokenization. Even with fewer tokens, our reconstructions maintain reasonable global alignment with ground truth, with an expected loss in detail.

## 2 RELATED WORK

The goal of image tokenization is to map high-dimensional 2D images into compressed latent representations. Modern vision systems often use self-supervised learning objectives, such as contrastive learning, self-distillation, in-painting, and generative modeling, to achieve this. Architecturally, these methods typically convert images into 2D feature embeddings using convolutional backbones or transformers after splitting images into patches. However, this approach constrains and tightly binds the processing, compression, and representation capacities to a **fixed number of processing units, down-sampling steps or patches, regardless of the input image.** Several prior works have explored such issues with different motivations, as discuss below –

**Dynamic Token Processing:** Several works (Bolya et al., 2023; Rao et al., 2021; Yin et al., 2022) focus on dynamically processing tokens in the ViT architecture by pruning or merging them across layers. Token Merging (Bolya et al., 2023) accelerates ViT by merging a fixed number of tokens per layer, resulting in a consistent token count for each image. Inspired by ACT (Graves, 2016) and Universal Transformers (Dehghani et al., 2018), DynamicVIT (Rao et al., 2021) and A-ViT (Yin et al., 2022) adaptively prune tokens or dynamically halt processing for different tokens, with a focus on classification tasks. Concurrent work (Jain et al., 2024) extends this by routing 2D image tokens through different experts, rather than pruning or merging, for classification and image retrieval. Tokens in these works remain tightly coupled to image patches. Our approach also involves dynamic token processing but primarily focuses on distilling images into a variable-length compressed 1D latent space via a self-supervised reconstruction objective, allowing each image to have flexible representational capacity beyond patch-based tokens.

**Flexible or Variable-Length Representation Learning:** Each image has varying levels of detail, making a single patch size insufficient for vision transformers. FlexViT (Beyer et al., 2022) uses variable patch sizes for multiple image representations, though theoretically its capacity is still limited by the smallest patch size and 2D token-patch bias. Matryoshka Representation Learning (Kusupati et al., 2022) learns flexible but fixed representations (bounded by the feature dimension) by ensuring low-dimensional subsets of a feature vector can perform classification and image retrieval. Concurrent works (Hu et al., 2024; Cai et al., 2024) extend this to token space, enforcing subsets of tokens to support vision-language tasks. We also learn variable-length token representations, but *unlike static Matryoshka methods, which learn all the representations in one go, we focus on recurrent processing & adaptive memory—iteratively refining & adding new latent tokens—opening doors for longer representations (for streaming data) through longer rollouts in future.* Recently published on arXiv, ElasticTok (Yan et al., 2024)—similar to Matryoshka representations—learns variable-length encodings for images and videos by learning a fixed, max-sized representation in one step, then searching for a mask to sample a subset of this full-length representation.

**Latent Tokens or 1D Tokenization:** To overcome the 2D token-patch bias, Perceiver (Jaegle et al., 2021b;a) distills 2D image tokens into 1D latent tokens not tied to specific patches, aiming for modality-agnostic transformers. Similar approaches, such as Recurrent Interface Networks (RIN) (Jabri et al., 2023), AdaTape (Xue et al., 2023), and Titok (Yu et al., 2024), perform 2D-to-1D distillation or read-write operations for generation, recognition, and reconstruction, respectively. RIN and Titok use fixed-length token representations, while AdaTape allows a one-time selection of

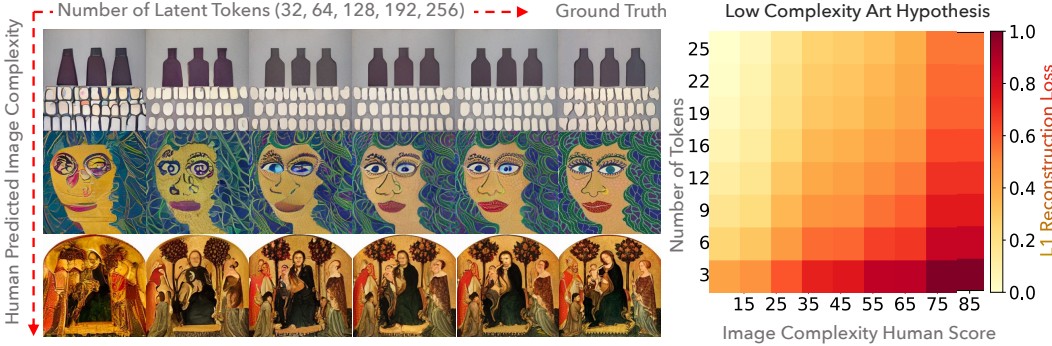

Figure 3: **Compression vs. Information Entropy Hypothesis on the Out-of-Distribution People-Art Dataset:** Adaptive tokenization enables analysis of the Low-Complexity Art Hypothesis by examining token requirements for images of varying complexity. The plot on the right clearly shows that as (human-annotated) **image complexity increases, so does the need for more computational tokens.** More complex images have higher L1 reconstruction loss at fewer token count.

variable-length latent tokens per input image. RIN uniquely performs recurrent read-write between image and latent tokens. Unlike these, we integrate 2D-to-1D distillation with both recurrent processing and adaptive memory—iteratively refining image and existing latent tokens, and adaptively adding more latent tokens as new memory. We also enable optional dynamic halting for improved distillation in regions needing further refinement. Beyond the technical differences, we emphasize that each image requires unique representation depending on complexity, familiarity, downstream model/task (Sec. 4). Moreover, the proposed recurrent computing with adaptive memory promotes emergent token specialization for object/part discovery, Fig. 7, Fig. 8, Appendix Fig. 15 and Fig. 16

**Relevant works on large-language models** include (Goyal et al., 2024; Herel & Mikolov, 2024) which allocate additional compute budget in terms of fixed additional "thinking" tokens, separate from input tokens. They demonstrate improved reasoning capabilities w/ thinking tokens for LLMs.

## 3 ADAPTIVE LENGTH IMAGE TOKENIZATION

Tokenization refers to the process of breaking down input data into discrete units or tokens, that are suitable for a specific downstream task. General-purpose tokenizers are usually trained with self-supervised objectives such as auto-encoding, next-token prediction, contrastive learning, or self-distillation. In the visual domain, prominent tokenizers like VAEs (Kingma & Welling, 2022), VQ-GAN (Esser et al., 2020), and ViT (Dosovitskiy et al., 2020) rely heavily on the 2D spatial inductive bias of images, treating 2D patches as tokens. This approach ties the tokenizer's architecture closely to the inductive bias of the visual domain and limits its representational capacity to a fixed number of tokens based on the number of image patches. The Perceiver line of research (Jaegle et al., 2021b;a) overcomes the 2D inductive bias limitation, (proposing a modality-agnositic architecture) while *still* distilling 2D image tokens into a *fixed* one-dimensional learned representation. In this work, we argue that **each image is unique and warrants different number of tokens**. To address this, we propose a novel framework that auto-regressively allocates more representational capacity (i.e., tokens) to an image, allowing for a variable number of tokens for each image at test time. We first outline the core auto-encoding modules — latent-distillation encoder and decoder — that distill 2D images into 1D tokens and back, and then introduce our approach of auto-regressive token allocation per image. Fig. 1 provides an overview of Adaptive Length Image Tokenizer (ALIT).

**Latent Distillation of 2D Image Tokens to 1D Tokens:** We want to map an input image to 1D latent tokens. Focusing on the core problem of compressive / adaptive representation learning (and primarily for compute reasons), we first leverage an existing VQGAN image tokenizer to first map an input image to a set of 2D image tokens, $\mathbf{K}_{2D}^{t=0}$. Credited to years of research done on quantized 2D auto-encoders, the pre-trained VQGAN model can map a $256 \times 256$ image to $16 \times 16$ 2D spatial tokens, without much loss of detail. Each of the $16 \times 16$ tokens is a pointer to one of the quantized codes in the trained VQGAN codebook. In this section, we distill the $\mathbf{K}_{2D}$ (= 256 for 256-dimensional image) spatial tokens to a few $\mathbf{K}_{1D}(\ll 256)$ 1D tokens. For majority of the experiments, we set this atomic (min. token count per image) number, $\mathbf{K}_{1D}$ to 32, for ease of experimentation.

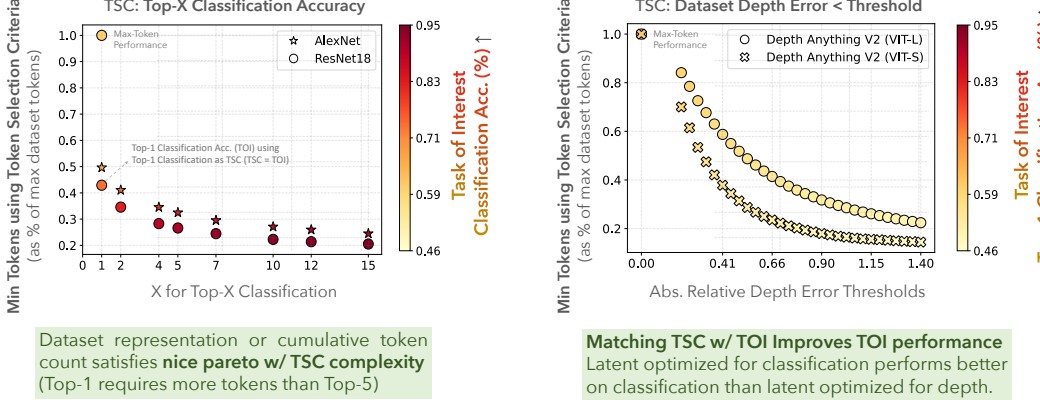

Figure 4: **Analysing Dataset Representation Capacity**: We vary tokens per image using different **Token Selection Criteria (TSC)** [2]– Best Top-X Classification Accuracy (Left) and Depth Error $<$ Threshold (Right). We use GT class/depth maps for computing TSC classification/depth errors. We then evaluate Classification Accuracy (**Task of Interest, TOI**) on the dataset reconstructed using different TSCs. X-axis = TSC, Y-axis = Dataset Token Count, Marker-Color = TOI Perf.

**2D→1D→2D Distillation** — Given $\mathbf{K}_{2D}$ spatial image tokens / features, each of dimension $\mathbf{d}_{2D}$, we append them with $\overline{\mathbf{K}}_{1D}$ latent tokens along the token axis, and pass then through the latent-distillation-encoder, $\mathbf{Enc}$. $\overline{\mathbf{K}}_{1D}$ are initialized with learned embeddings. The distillation encoder performs joint self-attention on all the tokens and distills $\mathbf{K_{2D}} \to \mathbf{K_{1D}}$. Although previous works (Jaegle et al., 2021b; Jabri et al., 2022) have experimented with cross-attention, we do not focus on this aspect of the architecture for this work and leave it for future analysis. The distilled latent tokens, $\mathbf{K}_{1D}$, are then passed to the distillation decoder $\mathbf{Dec}$, which appends them to $\overline{\mathbf{M}}_{2D}$ masked tokens and performs the reverse task of distilling the latent tokens back to the 2D spatial tokens i.e $\mathbf{K_{1D}} \to \mathbf{M_{2D}}$. All inputs to the distillation-encoder and distillation-decoder masked tokens are added with positional encoding (separate ones for 2D image tokens, 1D latent tokens and 2D masked tokens). We factorize and quantize the distilled latent tokens (output of $\mathbf{Enc}$) before passing them to the distillation-decoder, by sampling from a learned 1D codebook via closest codebook logic, following (Yu et al., 2021; 2024). Among other techniques (Zhu et al., 2024; Huh et al., 2023), we found factorization to be most useful for learning quantized 1D codebook.

$$\left.\begin{array}{r} \mathbf{K}_{2D}^{t=1}, \mathbf{K}_{1D}^{t=1} = \mathbf{Enc}\big(\big[\ \mathbf{K}_{2D}^{t=0}\ ;\ \overline{\mathbf{K}}_{1D}\ \big]\big) \\ \mathbf{M}_{2D}^{t=1} = \mathbf{Dec}\big(\big[\ \overline{\mathbf{M}}_{2D}\ ;\ \mathbf{K}_{1D}^{t=1}\ \big]\big) \end{array}\right\} \quad \text{Latent Distillation } 1^{st} \text{ Iteration}$$

$\overline{\mathbf{K}}_{1D} \to \mathbf{K}_{1D}$ denotes an encoder update to map initialized latent embedding to learned distilled embedding. Likewise, $\overline{\mathbf{M}}_{2D} \to \mathbf{M}_{2D}$ denotes reverse distillation using learned latent tokens ($\mathbf{K}_{1D}$) to map masked 2D tokens to reconstructed image tokens. t=0 to t=1 denotes one encoder update. The main learning objective is reconstruction loss between $\mathbf{M}_{2D}$ and $\mathbf{K}_{2D}^{t=0}$. [; ] denotes concatenation.

**Auto-regressive Framework for Variable Tokenization:** In the previous section, we explained the core module for 2D→1D distillation module. We now describe the **auto-regressive rolling of the encoder-decoder distillation module for learning variable tokens per image**, with $\mathbf{K}_{1D}^{t=1}$ as the minimum tokens per image. Multiple works (Dehghani et al., 2018; Graves, 2016) in sequential decision making and natural language processing perform recursive roll-out of the *same thinking* (Schwarzschild et al., 2021) architecture to provide more computational budget to the input task. In a similar vein, we perform recurrent processing of the input image with the objective of learning variable-length compressed representations. With each roll-out iteration, we not only provide more processing capacity by recursively rolling out the distillation $\mathbf{Enc} - \mathbf{Dec}$ architecture, but also provide *additional computational memory in terms of new writeable tokens to better distill image tokens into more 1D latents*. We now dive into the details.

---

[2]**Token-Selection Criteria (TSC) Example** – Selecting per-image tokens using TSC = Top-X Classification means – identifying the minimum number of tokens for each image such that the corresponding reconstructed image is correctly classified (by comparing against GT label) among the Top X predictions by ResNet-18.

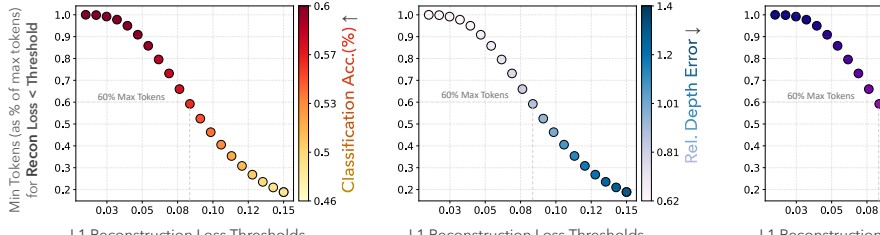

Figure 5: **TSC–TOI Alignment determines Dataset Representation Capacity**: We vary tokens per image via **"Reconstruction Loss $<$ Threshold" as automatic Token Selection Criteria (TSC)** and evaluate multiple Tasks of Interest (TOI) on the reconstructions. Since TSC determines dataset token count, *strong TSC–TOI alignment enables desired TOI performance at compressed representations — 60% of max-tokens (selected via Recon. TSC) achieve similar perf. on all TOIs as max tokens perf., supporting adaptive-tokenization.* Fig. 13 / 14 show classification / depth TSC plots.

At each iteration of latent distillation, we concatenate the latent tokens from the previous iteration, $\mathbf{K}_{1D}^{t=T}$, with additional newly initialized tokens (initialized with learned embeddings), $\overline{\mathbf{K}}_{1D}$. Optionally, to help the distillation encoder focus on image tokens that *were not perfectly distilled in the previous iteration*, we apply a masking / dynamic halting operation ($\mathbf{Mask}$) to the processed image tokens from the last iteration, $\mathbf{K}_{2D}^{t=T}$. This mask is determined by the alignment between reconstructed output $\mathbf{M}_{2D}^{t=T}$ and original image tokens $\mathbf{K}_{2D}^{t=0}$. The masked image tokens are then concatenated with the latent tokens and passed through the distillation encoder-decoder, $\mathbf{Enc} - \mathbf{Dec}$. This process is repeated across multiple iterations. As in single-step distillation, the primary training objective is to minimize the reconstruction loss between the new reconstruction, $\mathbf{M}_{2D}^{t=T+1}$, and the original image tokens $\mathbf{K}_{2D}^{t=0}$. At each iteration, the distilled latent tokens are factorized and quantized using a shared 1D codebook — tokens learned across different iterations belong to the same embedding space.

$$
\left.
\begin{aligned}
\mathbf{K}_{1D}^{t=T} &= \left[\ \mathbf{K}_{1D}^{t=T}\ ;\ \overline{\mathbf{K}}_{1D}\ \right] \\
\mathbf{K}_{2D}^{t=T} &= \mathbf{Mask}\left(\ \mathbf{K}_{2D}^{t=T}\ \mid\ \mathbf{M}_{2D}^{t=T}, \mathbf{K}_{2D}^{t=0}\ \right) \\
\mathbf{K}_{2D}^{t=T+1}, \mathbf{K}_{1D}^{t=T+1} &= \mathbf{Enc}\left(\left[\ \mathbf{K}_{2D}^{t=T}\ ;\ \mathbf{K}_{1D}^{t=T}\ \right]\right) \\
\mathbf{M}_{2D}^{t=T+1} &= \mathbf{Dec}\left(\left[\ \overline{\mathbf{M}}_{2D}\ ;\ \mathbf{K}_{1D}^{t=T+1}\ \right]\right)
\end{aligned}
\right\} \quad \text{Latent Distillation } \mathrm{T} + 1^{th} \text{ Iteration}
$$

In summary, at each iteration of the recurrent rollout, the latent tokens from the previous iteration receive residual updates, while new computational memory (additional latent tokens) is introduced. These new tokens give the existing latent tokens the freedom to focus on specialized regions, leading to sharper & sparser attention, as shown in Fig. 7, Fig. 8, Appendix Fig. 15 and Fig. 16.

**Training Procedure:** Our training follows a multi-stage approach similar to (Esser et al., 2020; Yu et al., 2024). We begin by reconstructing pre-trained VQGAN image tokens from distilled latent tokens, using smooth cross-entropy loss on the discrete VQGAN tokens (as explained above). Next, we jointly optimize VQGAN encoder-decoder and the distillation $\mathbf{Enc} - \mathbf{Dec}$ with an objective to reconstruct image pixels. Following (Esser et al., 2020), we introduce GAN loss later in the training process, once the image reconstruction quality has reached a satisfactory level across multiple representations, to further improve photo-realism. In addition to recon. and adversarial losses, we apply quantization losses throughout the training to facilitate the learning of 1D latent codebook. Please refer to Appendix Sec. A.6 for more details on the training procedure & implementation details.

## 4   NOT ALL IMAGES ARE WORTH THE SAME REPRESENTATION

*Each image is unique and requires a different number of tokens as representation.* Additionally, *each image or observation can have multiple valid representations*, echoing Epicurus' notion of multiple explanations. By mapping an image to various quantized latent spaces, the model learns to sample different tokens from the training set's codebook, optimizing the reconstruction objective at different levels of computational capacity. In this section, we provide experimental insights into how learning adaptive representations can help support and expand upon these concepts.

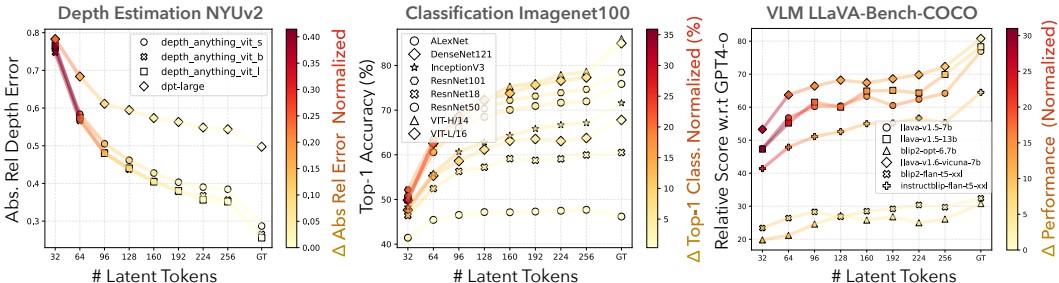

Figure 6: **Tokens vs. Model Strength:** Stronger models outperform weaker ones across all token counts but are more sensitive to fewer-token reconstructions, showing a sharper perf. drop (∼color) at fewer tokens. In contrast, **weaker models (lower GT perf.) can manage with fewer tokens.**

**Representation Capacity or Compression Depends on Information Entropy / Complexity:**
Schmidhuber's Low Complexity Art theory (Schmidhuber, 1996) correlates an image's perceptual complexity with its compressibility—the smallest description length of an image often aligns with its complexity. Given that our approach generates multiple compressed representations for a given image, we evaluate such correlation between human-labeled complexity estimates (ranging from 0 to 100) (Saraee et al., 2018) and the L1 reconstruction loss using our adaptive tokenizer at varying token capacities. Fig. 3 (left) illustrates reconstructions of completely out-of-distribution (OOD) images from the PeopleART dataset (Westlake et al., 2016), using between 32 and 256 tokens for images of different complexities. These reconstructions are produced using a model trained on ImageNet-100, which contains no art-related images and is significantly different from PeopleART. As seen, the low-complexity image (top row) is adequately reconstructed with fewer tokens, while the highly complex image (bottom row) requires more tokens for accurate reconstruction. Furthermore, the complexity-reconstruction correlation plot in Fig. 3 (right) perfectly highlights two observations: (a) *as image complexity increases, reconstructions with fewer tokens result in higher L1 errors, necessitating a larger memory budget*; and (b) at a fixed image complexity, increasing the computational budget (i.e., number of tokens) reduces the loss, *demonstrating the efficiency of the adaptive representation model*. See Appendix Fig. 11 and Fig. 12 for more such results.

**Representation Capacity Depends on Familiarity with the Training Set:** Similar to how out-of-syllabus questions require more effort, reconstructing OOD images demands more computational tokens. By learning quantized adaptive representations, our model maps test images to adaptive tokens by sampling from a learned trainset codebook. *This enables us to distinguish between in-distribution (IID) and out-of-distribution (OOD) images. IID images are more efficiently reconstructed with fewer tokens, as they can sample familiar representations from the trained codebook, whereas OOD images require more tokens.* From Tab. 1, the FID gap between 64 and 256 tokens is smallest on in-distribution ImageNet-100 validation set (7.92), larger on less in-distribution COCO dataset (12.56), and largest on highly OOD Wikipedia images (23.32). In summary, while all models exhibit performance loss on OOD images, the loss is most pronounced with fewer tokens. However, training larger adaptive tokenizers on larger datasets (eg: LAION) over longer periods may close the distribution gap, as evidenced by improved performance on IN-1K vs IN-100 (Fig. 9 third plot).

**Representational Capacity Depends on Downstream Task:** We utilize our variable-length representations to demonstrate how dataset representations vary across downstream tasks. To achieve this, we select different tokens for each image based on specific token-selection criteria (TSC) and analyze the minimum dataset representations required for optimal performance, plotting cumulative image token counts as a fraction of the total VQGAN tokens. By **dataset representation**, we mean cumulative token count for the dataset when different images are reconstructed using different token counts. For TSC=Classification [2], we evaluate the minimum tokens needed for the best top-1, top-2, ..., top-X accuracy. For TSC=Depth Estimation, we determine the token count necessary to achieve per-image relative depth errors below thresholds such as $0.2, 0.4, 0.6$. Likewise TSC=Reconstruction Loss, selects per image tokens based on L1 recon. loss $<$ certain thresholds. Notably – *Reconstruction Loss serves as an automatic (self-supervised) token selection criteria*, while Classification / Depth Error as TSC requires GT class-labels / Pseudo-GT depth maps.

In Fig. 4, we select minimum tokens based on two criteria: classification (left) and depth (right). The images reconstructed using selected tokens are then assessed using ResNet-18 for Classification Accuracy as the task-of-interest (TOI). *The resulting Pareto curves indicate that as token-selection*

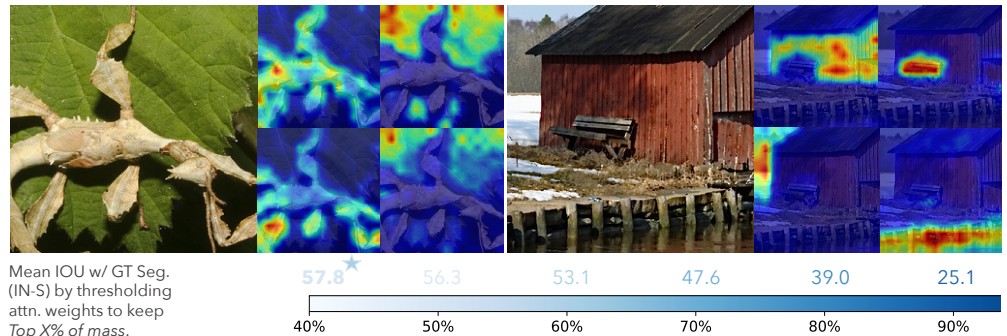

Figure 7: **Visualizing Latent Token Attention Maps:** The learned latent tokens effectively bind to distinct objects, suggesting potential for object discovery as future research. This contrasts with 2D tokenizers, where tokens are strongly biased to predefined patches (also see Appendix Fig. 15).

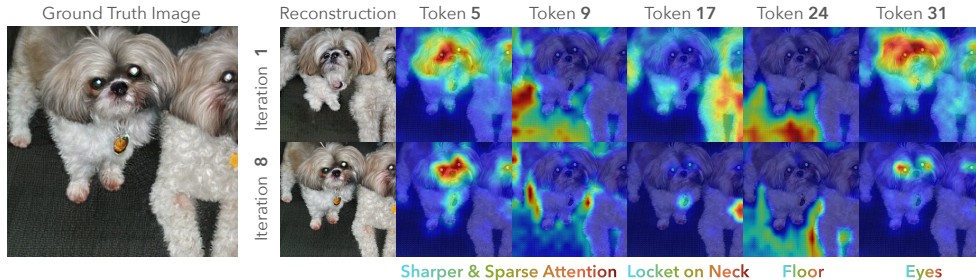

Figure 8: **Role of Recurrence on Latent Tokens:** Credited to increasing number of tokens in each iteration, the recurrent update on existing latent tokens makes them focus on sparser & more localized regions, leading to improved alignment w/ GT segmentation over iterations (see Tab. 6).

*criteria (TSC) increases in complexity, the representation capacity requirement also rises;* for example, achieving a depth error $< 0.41$ requires more tokens per image than a depth error $< 1.4$. Moreover, *optimal performance on task of interest occurs when tokens are selected using the same criteria as the downstream task (i.e. TSC=TOI, Best Top-1 Accuracy is achieved when both TSC and TOI are Classification (Fig. 4, left)*, compared to when TSC=Depth, TOI=Classification (Fig. 4, right)) and (b) selecting tokens based on arbitrary thresholds for depth or reconstruction loss has small impact on classification accuracy i.e. *classification accuracy with TSC = Depth Loss $< 0.41$ is similar to that with TSC = Depth Loss $< 1.40$*, suggesting that classification requires fewer tokens (as supported by Linear Probing Experiments in Sec. 5). Thus, optimal tokens per image depends on both token-selection criteria & the desired task.

Next, we explore the scenario where the token-selection criteria remain constant, but the task of interest (TOI) varies. Since the TSC enables sampling different dataset representations, *the optimal compression of dataset representations for maximum TOI performance depends on the alignment between the TSC and the TOI. In other words, dataset representations are more compressible when optimized and tested on a similar task.* This also aligns with the idea that representational capacity is influenced by familiarity with the training data. For instance, in Fig. 5, we allocate variable tokens per image using image reconstruction loss below a threshold as the token-selection criteria. The resulting token-reconstructions are then evaluated across three tasks: classification accuracy using ResNet-18, depth estimation accuracy using DepthAnythingV2, and FID using VGG16 features. *Approximately 60% of the total dataset representations (sampled using Reconstruction Loss as TSC) are sufficient to achieve near-optimal performance across all tasks—classification, depth estimation, and FID, highlighting that reconstruction loss could be a good self-supervised objective to select tokens per-image.* Appendix Fig. 14 and Fig. 13 analyze different Tasks of Interest (TOIs) performance using Depth Error Thresholding and Best Classification Accuracy as TSC respectively.

**Representational Capacity Depends on Model Strength or Current Knowledge**  Fig. 6 examines the relationship between downstream model strength and tokenized representations for tasks such as depth estimation, classification, and vision-language modeling. Weaker models exhibit *smaller performance drops with reduced token counts*, as shown by the color in Fig. 6, which repre-

| Approach | ImageNet100 | | | | | | | | COCO | | | Wikipedia (WIT) | | |
|---|---|---|---|---|---|---|---|---|---|---|---|---|---|---|
| | 32 | 64 | 96 | 128 | 160 | 192 | 224 | 256 | 32# / 64 | 128 | 256 | 32# / 64 | 128 | 256 |
| Titok-L-32 | 11.60 | - | - | - | - | - | - | - | 14.18# | - | - | 53.57# | - | - |
| Titok-B-64 | - | 8.22 | - | - | - | - | - | - | 9.15 | - | - | 42.86 | - | - |
| Titok-S-128 | - | - | - | 8.22 | - | - | - | - | - | 9.15 | - | - | 38.16 | - |
| VQ-GAN | - | - | - | - | - | - | - | 7.04 | - | - | 7.77 | - | - | 31.27 |
| Ours-S (IN1K) | 22.69 | 14.99 | 11.97 | 10.17 | 9.54 | 8.85 | 8.48 | 8.02 | - | - | - | - | - | - |
| Ours-S (IN100) | 22.57 | 16.17 | 13.30 | 11.69 | 10.22 | 9.30 | 8.55 | 8.25 | 22.28 | 14.22 | 9.72 | 61.77 | 47.91 | 38.45 |
| Ours-SemiLarge (IN100) | 19.70 | 13.92 | 11.39 | 10.41 | 9.23 | 8.75 | 8.22 | 8.03 | - | - | - | - | - | - |

Table 1: **Reconstruction FID ($\downarrow$) on different evaluation datasets** Our method performs comparably to VQGAN and Titok, despite being amortized over multiple iterations, allowing flexible representations. Longer training on larger datasets, w/ deeper n/w could bridge the gap (see Fig. 9 for scaling laws)[3]. The numbers below dataset titles are the variable token counts.

sents the perf. gap between GT and reconstructed images. For instance, fewer-token reconstructions have minimal impact on AlexNet (Classification) and the Blip model (vision-language modeling). Fig. 6 VLM experiments support concurrent works of Matryoshka VLMs (Hu et al., 2024; Cai et al., 2024), further promoting recurrent and adaptive tokenization using VLMs as TSC and TOI.

## 5 FURTHER EXPERIMENTS & ABLATIONS

In this section, we present additional experiments to evaluate our adaptive and recursive representations. We begin with image reconstruction experiments, followed by linear probing of learned representations for Top-1 classification. Next, we ablate the use of continuous vs. discrete tokenizers for reconstruction task and finally analyze the learned tokens, showcasing alignment with GT segmentation (with tokens hinting at object discovery), and the role of recurrent processing in achieving that. For implementation details, please refer to the Appendix Sec. A.6.

**Image Reconstruction:** We compare our adaptive tokenizer with the fixed-length 1D tokenizer Titok (Yu et al., 2024) and 2D tokenizer VQGAN (Esser et al., 2020) based on L1 reconstruction loss and FID scores on ImageNet100, COCO and Wikipedia Image-Text (WIT) datasets [3]. The reconstruction loss metrics and the number of images at different sampled reconstruction loss thresholds are shown in Fig. 2 (right plot). Unlike baselines with fixed-length representations, we *amortize the learning of variable length representations using a smaller network* – Titok-L-32 uses 24-layer encoder/decoder to generate 32 tokens, while Ours-S uses the same 8-layers to learn variable (32 to 256) tokens. Despite that, as shown in Fig. 2, *our method achieves slightly lower reconstruction loss compared to baselines, maintaining significant details of the input image, while the baselines, optimtized for realism with GAN loss, achieve slightly better FID than ours for low-token reconstructions*. Through the ablations shown in Fig. 9, we conjecture that multiple factors could enable further improving the slight FID loss (at low tokens) when training an amortized architecture for variable-length tokenization – larger models via both network depth and feature dimension (plot 1 & Fig. 17), training on larger dataset IN-1K vs IN-100 (plot 3 & row 5 vs row 6), longer training (plot 3), and most significantly GAN training (Sec. 3 Training Procedure).

**Linear Probing for ImageNet-1K Classification:** We evaluate the learned adaptive representations on the downstream task of classification by linear probing the latent representations, following standard practices (He et al., 2021). Specifically, we use the output of the latent-distillation encoder (mean pooling both the processed image tokens and 1D latent tokens). Our linear probe performance is on par with Titok models—Ours-S-32 (after 1st iteration) achieves 49.9% Top-1 Accuracy, compared to Titok-S-32's 48.0%, and Ours-SemiLarge achieves 59.5% compared to Titok-L-32's 60.0%, despite our model having a much smaller network (SemiLarge = 16 layers vs. L = 24 layers). Two key observations: (a) **Role of Recurrent Processing**—the single iteration output of Ours-L distillation encoder with 32 latent tokens achieves 55.6% Top-1 Accuracy, while recurrent processing through the encoder 2–3 times improves this to 59.5%. (b) **Distilling 2D tokens into 64 to 128 1D tokens is sufficient for the coarse classification task**; mean pool of first 64 to 128 latent tokens yield better linear-probing classification accuracy than pooling all (max. 256) latent tokens. Thus, the first few tokens capture most of the information required for classification, while additional tokens are crucial for reconstruction refinement. *Thus, recurrent processing sharpens/localizes token attention, improving both reconstruction and classification.*

---

[3]Unless mentioned, all experiments / visualizations were done using IN100 training of latent-distillation modules w/ base VQGAN/VAE tokenizer trained on IN1K. The results of the INK trained model are only in Tab. 1 and Appendix Tab. 2)

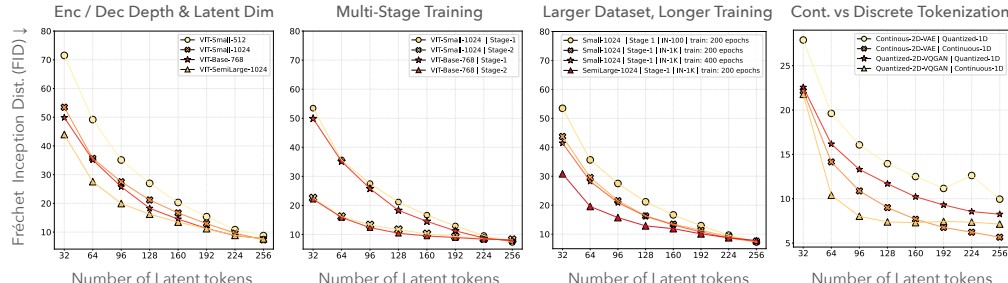

Figure 9: **(Plots 1-3)** Ablating FID improvements w/ larger n/w, longer training, larger datasets, and multi-stage GAN training. **(Plot 4)** Ablating quantized vs continuous 2D and 1D tokenizers.

**Continuous vs Discrete Tokenizer:** We learn adaptive representations by distilling a discrete VQ-GAN tokenizer into quantized 1D latents. Fig. 9 (plot 4), Appendix Tab. 7 and Appendix Fig. 18 compare continuous vs discrete tokenizers at both 2D image-token and 1D latent-token hierarchies. Using continuous 1D latents (i.e., no discrete quantization between distillation encoder/decoder) may improve FID more than a continuous 2D base tokenizer (both VAE and VQGAN with continuous 1D representations perform well). However, quantized 1D representations can achieve more compressed encodings ($32 \times 12$ bits per image for a 4096-size codebook) compared to continuous embeddings ($32 \times 12 \times 32$ bits). Additionally, quantization could potentially help distinguish OOD from IID test images (see Sec. 4, second point).

**Analyzing Learned Tokens for Object Discovery:** We analyze the attention maps of learned 1D latent tokens to 2D image tokens. Fig. 7 shows attention maps corresponding to four tokens from the $7^{th}$ layer of Ours-S distillation-decoder. Many latent tokens correspond to *semantically meaningful objects or parts, suggesting emergence of object discovery*. This contrasts with 2D tokenizers, where each token inductively maps to an image patch, and only single class token attention heads (Caron et al., 2021) hold such semantic information. To further investigate this, following DINO (Caron et al., 2021), we computed attention map alignment with ImageNet-S GT segmentation by thresholding the top X% of attention maps as emergent seg maps. With $40\%$ attention, we achieve $57.8$ mean IOU, and by adjusting the threshold per image, we reach $71.8$ mIOU. Notably, our tokens are not optimized for segmentation. Such high-alignment with segmentation is an emergent property.

**Ablating Recurrence on Latent Tokens:** We ablate the effect of recurrent updates on latent tokens. As outlined in Sec. 3, with each iteration of recurrent and adaptive tokenization, previous tokens receive residual updates while new tokens are added. Fig. 8 illustrates how the attention maps of the first 32 tokens evolve after 8 iterations. Initially, *under constrained memory/few token settings, these tokens attend broadly to reconstruct the image well, but as more memory/latent tokens are added, they focus on sparser, more meaningful regions.* This sharpening, as seen in Fig. 8 token numbers 17, 24, and 31, attending to the eyes, locket, and floor regions, demonstrates the impact of recurrence (see Appendix Fig. 15 and Fig. 16 for more interesting and dense visualizations).

## 6 CONCLUSION

In this work, we propose a variable-length tokenizer for 2D images that operates by recurrently processing 2D image tokens, distilling them into 1D latent tokens, and adaptively adding new 1D tokens as computational resources for future iterations. This recurrent processing and adaptive computation enable the learned latent tokens to correspond to semantically meaningful objects or parts in the input image. We demonstrate comparable performance on reconstruction metrics and ImageNet-1K linear-probing experiments. Finally, we utilize per-image learned adaptive representations (and cumulative dataset representations) to highlight alignment of the required image representational capacity with – information entropy, familiarity with train set, knowledge of downstream tasks/models.

**Future Work:** We believe that recurrently learning variable-length representations could open up intriguing directions, such as large-scale video representation learning or long-horizon video understanding, where simply learning fixed-length representations may not even be a feasible solution. Other interesting avenues include exploring adaptive and recurrent nature of the proposed tokenizer as test-time thinking units and training for tasks like vision-language & visual-abstract reasoning (e.g.: ARC (Chollet, 2019)). Variable-token compression could speed up generative models.

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

## A APPENDIX

### A.1 ACKNOWLEDGMENTS

This work was supported by the Defence Science and Technology Agency, Singapore, MIT/IBM Agreement No. W1771646, MIT/Hyundai Motor Company RD Center Agmt Dtd 2/22/23, NSF CIF 1955864 (Occlusion and Directional Resolution in Computational Imaging), NSF PHY-2019786 (The NSF AI Institute for Artificial Intelligence and Fundamental Interactions, http://iaifi.org/) and the Department of the Air Force Artificial Intelligence Accelerator and was accomplished under Cooperative Agreement Number FA8750-19-2-1000.

We would like to thank our lab members — Ayush, Congyue, Ishaan, Jyo, Kabir, Manel, Prafull, Shamit, Tianwei, Tianyuan, Yichen for feedback and support & Qihang Yu for questions regarding Titok baseline.

### A.2 LIST OF FIGURES

### A.3 LIST OF TABLES

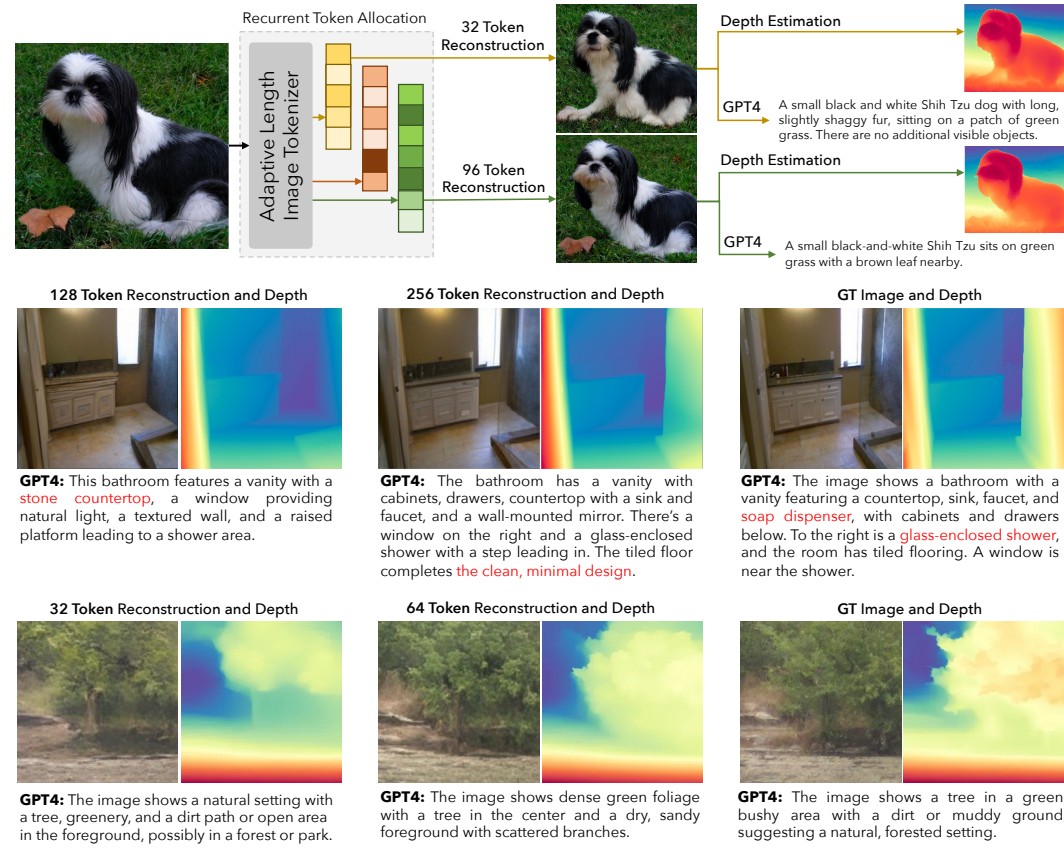

Figure 10: **Adaptive Length Image Tokenization** maps an image to multiple variable-length representations through a recurrent token allocation process, **enabling task-specific sampling**. We learn the tokenizer via image reconstruction as a self-supervised objective. While a compressed representation can be optimized for specific tasks (e.g., fewer tokens for "dog", "leaf", "grass" may suffice for a VLM task), reconstruction objective supports learning a universal, task-agnostic tokenizer.

## A.4    ADDITIONAL EXPERIMENTAL RESULTS

**Low-Complexity Art Hypothesis or Representation Capacity ~ Image Entropy**    Here, we provide additional experimental analysis supporting *representation capacity alignment with image entropy* claim made in the main paper, Sec. 4. We extracted human-annotated image complexity scores for the MSCOCO and Places2 subsets of the SAVOIAS dataset (Saraee et al., 2018). We leverage a model trained on ImageNet-100 to study this hypothesis, though both COCO and Places may be slightly out-of-distribution (OOD) relative to the learned codebook samples. The complexity-reconstruction loss alignment plots in Fig. 12 and Fig. 11 reaffirm the analysis in the paper—an increase in image complexity demands more tokens, and more tokens lead to smaller recon. loss.

**Representational capacity depends on downstream tasks:**    In addition to Fig. 5 in the main paper, where we evaluated Reconstruction Loss Thresholding as a Token Selection Criterion (TSC), we further assess GT-oracle-based Classification and pseudo-GT-oracle-based Depth Error metrics as TSC through Fig. 13 and Fig. 14. For Top-X Classification as TSC, an image receives 32 latent tokens if ResNet-18's top-X predictions from the 32-token reconstruction include the ground truth class. **Key Takeaways** – Compared to depth, reconstruction loss, and FID as the tasks of interest (TOI), the decrease in classification accuracy with a reduction in dataset representation capacity (cumulative tokens per dataset) is relatively smaller due to classification being a coarser task. When TSC $\neq$ TOI, using different TSCs to sample *the same fraction of tokens yields similar TOI performance*. For example, FID and Reconstruction Loss at 40% of max token capacity are comparable (compare $2^{nd}, 3^{rd}$ plots of Fig. 13 and Fig. 14), regardless of whether depth or classification was the token-selection criterion. However, compared to Top-X classification, choosing the more granular tasks of depth and reconstruction as TSC provide finer support for token sampling – therefore better/ diverse performance range for the TOI. Reconstruction loss serves as self-supervised TSC.

**Comparison with Flexible Embedding Baselines:** At the time of submission, Matryoshka (Kusupati et al., 2022) and FlexVIT-based (Beyer et al., 2022) approaches had only been explored for classification. The application of Matryoshka to reconstruction was a concurrent ICLR 2025 work (Yan et al., 2024). We implemented FlexVIT for reconstruction by applying variable patching on VQGAN 2D image tokens and designed a Matryoshka-style model by selecting a variable subset (either 32, 64, 128, or the full 256) of distilled 1D tokens during training with no recurrence. Tab. 4 compares ALIT with both approaches—our experiments indicate that both Matryoshka- and FlexVIT-based methods required larger parameter counts to match ALIT's performance. Fig. 21 visually compares the tokens learned by ALIT and the Matryoshka-based approach. Thanks to its recurrent nature, ALIT refines tokens over multiple iterations, leading to more specialized token bindings. Tab. 5 compares Matryoshka-style model with ALIT on OOD Coco and WIT datasets.

Finally, some more comparison between recurrent (ALIT) and masking-based (Matryoshka) approaches: (a) Compared to the Matryoshka approach, an ALIT model with the same number of parameters *utilizes more FLOPs at test time to learn longer representations due to its recurrent nature.* (b) In Matryoshka and other masking-based approaches, the 256-token representation always includes the same first 32 tokens as the 32-token representation, *which may not be optimal, as a higher token budget could allow for a more efficient utilization of those 32 tokens.* (c) Combining ALIT with both Matryoshka and FlexVIT approaches could yield a highly efficient tokenizer by leveraging the benefits of flexible 2D patches, adaptability at each recurrent step, and the ability to allocate more FLOPs through recurrence.

**Additional Ablations:** Tab. 3 and Tab. 7 specify the exact metrics used to generate the plots in Fig. 9. Both the plots and the tables clearly indicate that longer training, larger datasets (IN-100 vs. IN-1K), and multi-stage training are vital for achieving photo-realistic reconstructions. Furthermore, Fig. 17 qualitatively highlights the benefits of using larger models (ALIT-L vs ALIT-B vs ALIT-S) on image reconstruction quality. Fig. 18 visually analyses the use of continuous vs discrete tokenization at both 2D and 1D token levels. Continuous 1D Tokenization performs best on image-reconstruction task. That said, studying the benefits of quantization for retrieval and long-horizon tasks like visual abstract reasoning, video question answering would be useful. Furthermore, Tab. 10 and Tab. 11 ablate the role of 1D latent token codebook sharing and codebook size on reconstruction FID. Tab. 12 ablates dynamic halting procedure (see Fig. 1 or method section to understand where dynamic halting is performed). Finally, Tab. 8 and Tab. 9 describe the computational needs of ALIT in terms of inference time and gflops.

## A.5 ANALYSING THE LEARNED LATENT TOKENS

**Emergent Object Binding:** Compared to 2D tokenizers, where each token is bound to a patch (other than class or global tokens), the learned 1D tokens demonstrate signs of object binding and localization. In addition to Fig. 7 and Fig. 8, we showcase more examples of how the learned latent tokens bind to semantically meaningful objects on COCO dataset. For instance, in Fig. 15, note the token-zebra binding in the second example, the near-perfect segmentation of the bicycle symbol on the lane in row 3, the segmentation of the frisbee and human into different tokens in example 4, and the clustering of both plants (as indicated by high attention weights) in the final example.

**Comparison with DINO for Emergent Object Binding:** Fig. 19 visualizes the tokens learned by DINO and ALIT. Despite being trained with different objectives on different datasets, both representation learning systems exhibit intriguing emergent object discovery behavior. For DINO, the number of token-attention map visualizations is fixed and determined by the number of attention heads (=6 for DINO-S) multiplied by the number of class tokens (=1). Additionally, since DINO is trained with a self-distillation regime that matches intra-image crop representations, it is primarily optimized for classification. As a result, the class-token attention heads tend to focus on regions crucial for image classification (e.g., in Fig. 19, row 4, the tokens attend either to the human or the phone, aiding in "selfie" classification). In contrast, ALIT employs a variable number of tokens, leading to a flexible number of token-attention maps and enabling the discovery of multiple objects—notice, for instance, the switch segment detected on the background wall (row 4 of Fig. 19). We believe this strong emergent behavior arises from ALIT's reconstruction-based training objective, the discrete nature of its learned representation, and the recurrent refinement of its tokens.

**Recurrent Processing of Latent Tokens:** We further explore the role of recurrence in enhancing token attention and binding to semantic objects and parts. With each iteration of the model rollout, new tokens are added, allowing existing tokens to focus on specialized tasks rather than covering

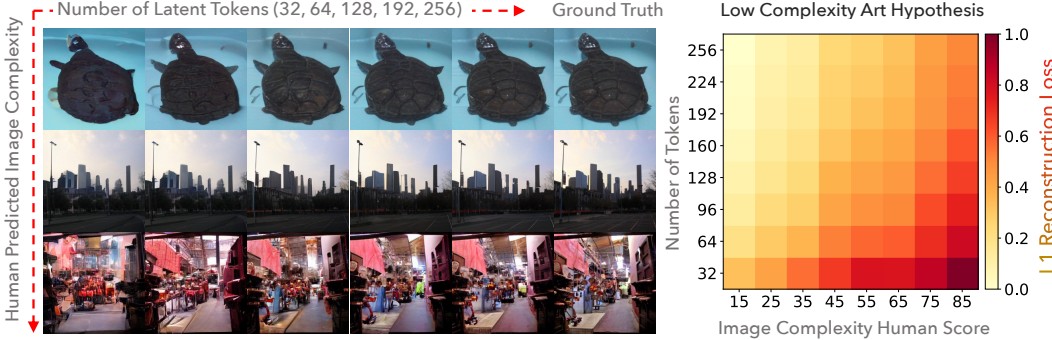

Figure 11: **Compression vs. Entropy Hypothesis on the Places Dataset:** Adaptive tokenization enables analysis of Low-Complexity Art Hypothesis by examining token requirements for images of varying complexity. The plot on the right clearly shows that as image complexity increases, so does the need for more computational tokens. These reconstructions use Imagenet100-trained model.

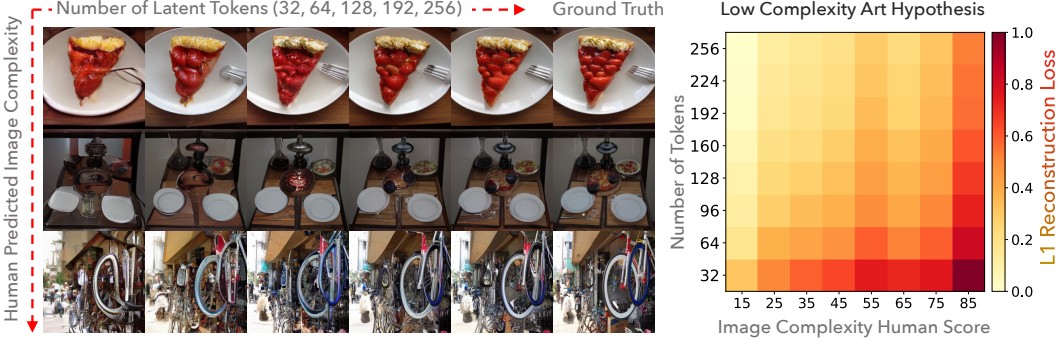

Figure 12: **Compression vs. Information Entropy Hypothesis on the MSCOCO Objects Dataset:** Adaptive tokenization enables analysis of Low-Complexity Art Hypothesis by examining token requirements for images of varying complexity. The plot on the right clearly shows that as image complexity increases, so does the need for more computational tokens.

the entire image. Figure 16 illustrates five examples from the COCO dataset, showing how the attention maps of selected tokens receive residual updates over eight recurrent iterations, leading to improved object binding. To quantitatively support this enhanced localization, we calculate mIOU segmentation metrics at different recurrent iterations in Tab. 6. These metrics reinforce the visual results, with mIOU consistently increasing across iterations at various attention masking percentiles.

**Analyzing the Role of Recurrence in 1D Codebook Utilization:** We visualize the frequency with which each code in the learned codebook is sampled across 5K validation images in Fig. 22. As shown in the figure, at later iterations, more tokens exhibit higher sampling frequencies (x-axis: token sampling frequency/count, y-axis: number of tokens at that frequency). This suggests that with more recurrent iterations, certain specialized tokens are sampled more frequently, reinforcing their role in capturing emergent object representations. This aligns with the token-object binding visualizations presented throughout the paper.

## A.6 EXPERIMENTAL DETAILS

**Training Details** In this section, we provide additional training details (see Sec. 3 for approach details and training procedure summary). We train ALIT in two phases – *latent-distillation pre-training and full fine-tuning stage (with gan loss)*. In the **latent-distillation pre-training stage**, we leverage a pre-trained image tokenizer (VQGAN or VAE) which maps an input image to 2D tokens. We only train the latent-distillation encoder-decoder modules in this stage, using image token reconstruction loss as the core-learning objective. With VQGAN as base tokenzier, we use cross-entropy loss comparing predicted logits with the ground-truth VQGAN-codebook index at each 2D token position. We use mean-squre reconstruction loss when using VAE as the base-tokenizer. We unroll the recurrent token allocation procedure for 8 iterations, expanding token memory from 32 (in $1^{st}$ iteration) to 256 (in $8^{th}$) during training. All the recurrent rollouts are trained end-to-end. At each iteration, we process the image-tokens, the existing 1D latent tokens

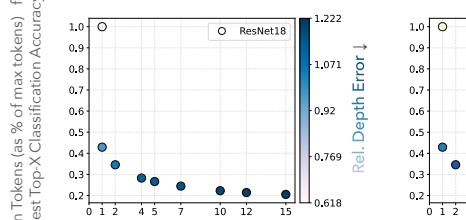 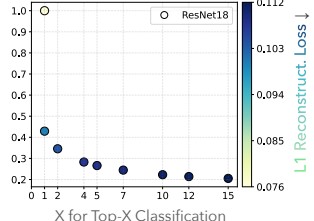 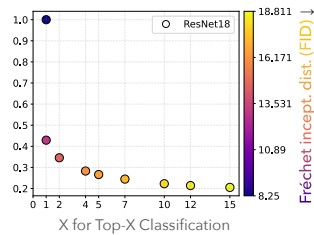

Figure 13: **Dataset Representation Analysis as factor of TSC-TOI Alignment**: We vary tokens per image using **"Best Top-X Classification" Accuracy as the Token Selection Criteria (TSC)** and evaluate different Tasks of Interest (TOI) on the reconstructions. We sample tokens per image such that each reconstructed image is classified correctly under Top-X Classification criteria, and then evaluate different downstream tasks of interests (TOIs) on the reconstructed dataset.

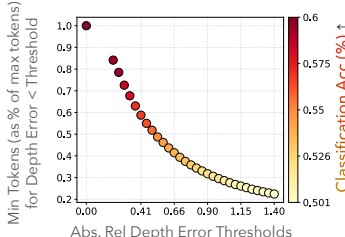 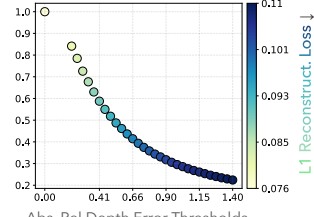 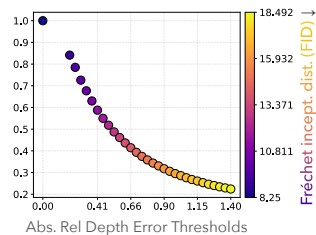

Figure 14: **Dataset Representation as a factor of TSC-TOI Alignment**: We vary tokens per image using **"Depth Loss < some Threshold" as the Token Selection Criteria (TSC)** and evaluate different Tasks of Interest (TOI) on the reconstructions. Being a dense task, depth error thresholding at several continuous thresholds provide more granular and diverse support to dataset representation compression and TOI performance, compared to Top-X Classification (X being discrete integer).

and add new latent tokens. During this training phase, we perform dynamic halting of the image tokens in each iteration, allowing the latent-distillation modules to focus on distilling image tokens which cannot be reconstructed perfectly till current iterations. We use transformer backbones for both latent-distillation encoder and decoder, performing self-attention among 2D image tokens and latent 1D tokens. In the **next training phase, we jointly fine-tune** both the base image tokenizer modules and the latent-distillation encoder-decoder modules with losses directly at the pixel-space. The training objectives are pixel-level reconstruction and adversarial losses (gan generator and discriminator losses) inspired from VQGAN (Esser et al., 2020) training procedure. We optimize for reconstruction loss for first few epochs, later switching to both reconstruction and adversarial losses. We recurrently map an image to variable-tokens (increasing token count by 32 in each iteration) for 8 iterations. However, unlike previous stage, we only compute loss at only iteration, thus we only need to perform recurrent processing of the latent-distillation encoder in each training run, executing the latent-distillation decoder and the base tokenizer decoder only once. This helps speed up the training and the compute memory requirements. No dynamic halting is performed in this phase. Thanks to the gan losses, this phase help boost the photo-realism / FID metric of the reconstructions, specially at lower-token counts. For more low-level details, please refer to our codebase – https://github.com/ShivamDuggal4/adaptive-length-tokenizer.

**Dataset** For training the adaptive tokenizer, we mainly utilize Imagenet-100 (100 classes of ImageNet-1K as used in (Wang & Isola, 2022)) and ImageNet-1K datasets. Fig. 9 showcase the scaling potential of training on larger datasets. For exact details of the 100 classes, please refer to our codebase. **Following are pointers to image sources used to create the paper figures and plots** — The three images shown in Fig. 10 are from ImageNet-100 valiation set (dog image), OOD indoor scene image is from NYUV2 dataset (Nathan Silberman & Fergus, 2012) and tree example is from WIT dataset (Srinivasan et al., 2021). Fig. 3, Fig. 11, Fig. 12 all leverage data from SAVOIAS dataset, which again is different from ImageNet-100 classes. Plots in Fig. 4, Fig. 5, Fig. 14, Fig. 13 all leverage ImageNet-100 validation set. Fig. 7 and Fig. 8 use ImageNet images for token visualzation, Fig. 15 and Fig. 16) use COCO images. Fig. 17 and Fig. 18 showcase randomly sampled images from ImageNet-100 validation set.

| Approach | ImageNet100 | | | | | | | |
| --- | --- | --- | --- | --- | --- | --- | --- | --- |
| | 32 | 64 | 96 | 128 | 160 | 192 | 224 | 256 |
| Titok-L-32 | 12.29 | - | - | - | - | - | - | - |
| Titok-B-64 | - | **9.72** | - | - | - | - | - | - |
| Titok-S-128 | - | - | 9.72 | - | - | - | - | - |
| VQ-GAN | - | - | - | - | - | - | - | 7.99 |
| Ours-S (IN1K) | **11.50** | 9.81 | **8.78** | **8.36** | **8.01** | **7.65** | **7.53** | **7.48** |
| Ours-S (IN100) | 11.77 | 10.13 | 9.26 | 8.77 | 8.32 | 7.95 | 7.81 | 7.62 |
| Ours-SemiLarge (IN100) | 11.81 | 10.09 | 9.16 | 8.62 | 8.36 | 8.01 | 7.79 | 7.63 |

Table 2: **Reconstruction L1 Loss** ($\times 100 \downarrow$) **evaluated on IN100 validation set.** For last two rows, our latent-distillation modules are only trained on IN100, rest everything is trained on IN1K.

| Train Dataset | Train Stage | Train Epochs | Enc./ Dec. Depth | Embed. Dim. | Number of Tokens | | | | | | | |
| --- | --- | --- | --- | --- | --- | --- | --- | --- | --- | --- | --- | --- |
| | | | | | 32 | 64 | 96 | 128 | 160 | 192 | 224 | 256 |
| IN-100 | Stage1 | 200 | small (8) | 512 | 71.53 | 49.18 | 35.11 | 26.97 | 20.31 | 15.38 | 10.84 | 8.77 |
| IN -100 | Stage1 | 200 | base (12) | 768 | 49.88 | 35.16 | 25.79 | 18.29 | 14.52 | 11.27 | 8.77 | 7.74 |
| IN-100 | Stage1 | 200 | semi-large (16) | 1024 | 43.97 | 27.56 | 19.94 | 16.22 | 13.48 | 11.19 | 8.86 | 7.64 |
| IN-100 | Stage1 | 200 | small (8) | 1024 | 53.47 | 35.66 | 27.51 | 21.18 | 16.66 | 12.94 | 9.66 | 7.29 |
| IN-100 | Stage2 | 200 | small (8) | 1024 | 22.57 | 16.17 | 13.30 | 11.69 | 10.22 | 9.30 | 8.55 | 8.25 |
| IN-100 | Stage2 | 200 | base (12) | 768 | 22.15 | 15.88 | 12.57 | 10.44 | 9.50 | 8.96 | 8.39 | 8.19 |
| IN-1K | Stage1 | 200 | small (8) | 1024 | 43.73 | 29.56 | 21.55 | 16.30 | 13.45 | 11.33 | 9.11 | 7.59 |
| IN-1K | Stage1 | 400 | small (8) | 1024 | 41.46 | 28.26 | 20.90 | 16.21 | 13.21 | 10.74 | 8.68 | 7.39 |
| IN-1K | Stage1 | 200 | semi-large (16) | 1024 | 30.84 | 19.56 | 15.72 | 12.87 | 11.90 | 10.11 | 8.76 | 7.78 |

Table 3: **Image Reconstruction Ablations (FID metric):** The ablations clearly highlights – longer training, larger dataset, larger network depth helps in improving FID. GAN base-training is critical.

| Approach | ImageNet100 | | | | | | | |
| --- | --- | --- | --- | --- | --- | --- | --- | --- |
| | 32 | 64 | 96 | 128 | 160 | 192 | 224 | 256 |
| Matryoshka Enc-S w/ ALIT Dec-S | 33.11 | 20.08 | 15.10 | 12.84 | 11.45 | 10.51 | 9.90 | 9.50 |
| Matryoshka Enc-SemiLarge w/ ALIT Dec-S[#] | 32.12 | 19.25 | 14.52 | 12.48 | 11.23 | 10.11 | 9.48 | 9.14 |
| **Matryoshka Enc-SemiLarge w/ ALIT Dec-SemiLarge** | **22.85** | **14.51** | **11.63** | **10.13** | **9.23** | **8.83** | **8.42** | **8.11** |
| FlexVIT Enc-S w/ ALIT Dec-S | 79.84 | 39.73 | - | 15.16 | - | - | - | 12.64 |
| FlexVIT Enc-SemiLarge w/ ALIT Dec-S | 25.62 | 15.77 | - | 10.21 | - | - | - | 7.90 |
| **FlexVIT Enc-SemiLarge w/ ALIT Dec-SemiLarge** | **23.30** | **14.73** | - | **9.92** | - | - | - | **7.80** |
| Ours-S | 22.57 | 16.17 | 13.30 | 11.69 | 10.22 | 9.30 | 8.55 | 8.25 |
| **Ours-SemiLarge** | **19.70** | **13.92** | **11.39** | **10.41** | **9.23** | **8.75** | **8.22** | **8.03** |

Table 4: **Reconstruction FID** ($\downarrow$) **comparison on ImageNet100:** Matryoshka and FlexVIT were originally proposed for classification tasks. We adapt them for dense reconstruction by employing ALIT-style 1D token decoding. We believe that combining Matryoshka-style masking and FlexVIT-style variable 2D patch-size approaches with the ALIT-style recurrent method could yield further benefits. (# indicates that the GAN discriminator runs 4 times more than the generator).

| Approach | COCO | | | Wikipedia (WIT) | | |
| --- | --- | --- | --- | --- | --- | --- |
| | 64 Tokens | 128 Tokens | 256 Tokens | 64 Tokens | 128 Tokens | 256 Tokens |
| Matryoshka-L | 23.05 | 14.43 | 10.66 | 62.02 | 48.21 | 40.28 |
| Ours-S | 22.28 | 14.22 | 9.72 | 61.77 | 47.91 | **38.45** |
| Ours-L | **21.44** | **13.64** | **9.71** | **60.11** | **47.52** | 38.69 |

Table 5: **Reconstruction FID** ($\downarrow$) **on COCO and Wikipedia Image-Text (WIT)**. All models are trained on Imagenet100. Our method (Ours-S, Ours-L) outperforms Matryoshka-L on the OOD datasets. Here "L" = "Semi-Large".

| Attention Threshold. Percentile | Recurrent Iteration | | | | | | | |
| --- | --- | --- | --- | --- | --- | --- | --- | --- |
| | 1 | 2 | 3 | 4 | 5 | 6 | 7 | 8 |
| 40% | 56.7 | 57.4 | 57.5 | 57.6 | 57.6 | 57.7 | 57.7 | **57.8** |
| 60% | 51.5 | 52.4 | 52.5 | 52.7 | 52.8 | 52.9 | 53.0 | **53.1** |
| 80% | 37.1 | 38.6 | 38.4 | 38.6 | 38.7 | 38.8 | 39.0 | **39.0** |

Table 6: **Recurrent Processing enhances Alignment of Token Attn Maps with GT Segments.** *Note – we did not optimize for segmentation, and token attention maps binding to segments is an emergent property.* This binding improves with recurrent processing (under increasing memory constraint). To extract a segment from a token attn map, we threshold the top X% of the attn map.

| Base 2D Tokenizer | Latent Dimension | Latent Quantization | Number of Tokens | | | | | | | |
|---|---|---|---|---|---|---|---|---|---|---|
| | | | 32 | 64 | 96 | 128 | 160 | 192 | 224 | 256 |
| VAE | 12 | ✓ | 27.90 | 19.62 | 16.07 | 13.95 | 12.50 | 11.16 | 12.62 | 9.95 |
| VAE | 12 | × | 22.14 | 14.15 | 10.88 | 9.00 | 7.68 | 6.76 | 6.22 | 5.66 |
| VQGAN | 12 | ✓ | 22.57 | 16.17 | 13.30 | 11.69 | 10.22 | 9.30 | 8.55 | 8.25 |
| VQGAN | 12 | × | 21.77 | 10.39 | 8.02 | 7.38 | 7.30 | 7.44 | 7.36 | 7.14 |

Table 7: **Continuous vs. Discrete Tokenization:** Continuous 1D latents (without quantization) are advantageous for image reconstruction, while quantization provides the benefit of compressed encoding. All token-object binding visualizations are performed using the VQGAN + latent quantization approach, showcasing the advantages of quantization beyond compression and error bounding.

| Approach | Number of Iterations | | | | | | | |
|---|---|---|---|---|---|---|---|---|
| | Iter 1 | Iter 2 | Iter 3 | Iter 4 | Iter 5 | Iter 6 | Iter 7 | Iter 8 |
| Titok-L-32 | 9.54 | - | - | - | - | - | - | - |
| Ours-S | 6.84 | 10.40 | 14.13 | 18.04 | 22.03 | 25.99 | 30.20 | 34.74 |

Table 8: **Iteration-wise Inference Time (ms) Comparison** on a single H100 GPU with FP32 precision for single-image encoding. The overhead of running an additional iteration of the adaptive tokenizer encoder is actually quite small: 4ms run time and $< +30$ gflops on a single H100 gpu. Overall, +30 gflops overhead per iteration is very less for modern compute and can be further reduced by using techniques like KV caching previous layer attentions etc.

| Approach | Number of Iterations | | | | | | | |
|---|---|---|---|---|---|---|---|---|
| | Iter 1 | Iter 2 | Iter 3 | Iter 4 | Iter 5 | Iter 6 | Iter 7 | Iter 8 |
| Titok-L-32 | 58.44 | - | - | - | - | - | - | - |
| Ours-S | 83.88 | 105.45 | 129.16 | 155.03 | 183.04 | 213.21 | 245.53 | 278.00 |

Table 9: **GFlops Comparison on a single H100 GPU for single-image encoding.** For 32 tokens, we our latent-distillation encoder has 8 transformer layers much smaller than Titok-L-32 (with same number of attention heads, feature dim and same number of tokens). Despite that, we have more GFlops because of using VQGAN image encoder + a small additional patch embedding layer as the image 2D embedding. We can switch to Titok-L-32 style image encoding and reduce upto 50 gflops.

| Codebook Size | Codebook Sharing | Number of tokens | | | | | | | |
|---|---|---|---|---|---|---|---|---|---|
| | | 32 | 64 | 96 | 128 | 160 | 192 | 224 | 256 |
| 4096 | × | 26.84 | 18.71 | 14.71 | 12.82 | 10.91 | 10.02 | 9.05 | 8.61 |
| 4096 | ✓ | **22.57** | **16.17** | **13.30** | **11.69** | **10.22** | **9.30** | **8.55** | **8.25** |

Table 10: **Effect of Codebook Sharing on Reconstruction FID** ($\downarrow$). Shared codebook leads to (a) more compression in terms of total codes. (b) theoretically allows one to rollout the adaptive tokenizer for much longer rollouts than done during training by having timestep-invariant codebook.

| Codebook Size | Number of tokens | | | | | | | |
|---|---|---|---|---|---|---|---|---|
| | 32 | 64 | 96 | 128 | 160 | 192 | 224 | 256 |
| 2048 | 26.65 | 18.91 | 14.76 | 12.20 | 10.86 | 10.03 | 9.49 | 9.13 |
| 4096 | **22.57** | **16.17** | 13.30 | 11.69 | 10.22 | 9.30 | **8.55** | **8.25** |
| 8196 | 23.76 | 16.55 | **13.11** | **11.14** | **9.75** | **9.17** | 8.61 | 8.42 |

Table 11: **Reconstruction FID vs 1D Codebook Size** There is not much (marginal) difference in performance between 4096 and 8196 codebook size, stating that the tokenizer is less sensitive to the codebook size. That said, when scaling to large models, scaling up the codebook while maintaining utilization would be an intruiging research question. 4096 is a usual codebook size for Imagenet.

| Dynamic Halting | Number of tokens | | | | | | | |
|---|---|---|---|---|---|---|---|---|
| | 32 | 64 | 96 | 128 | 160 | 192 | 224 | 256 |
| × | 24.38 | 17.00 | 13.49 | **11.53** | 10.30 | 9.43 | 8.76 | 8.29 |
| ✓ | **22.57** | **16.17** | **13.30** | 11.69 | **10.22** | **9.30** | **8.55** | **8.25** |

Table 12: **Ablating Dynamic Halting for Reconstruction FID metric.** Dynamic Halting is done only in stage 1 of training (and is optional).

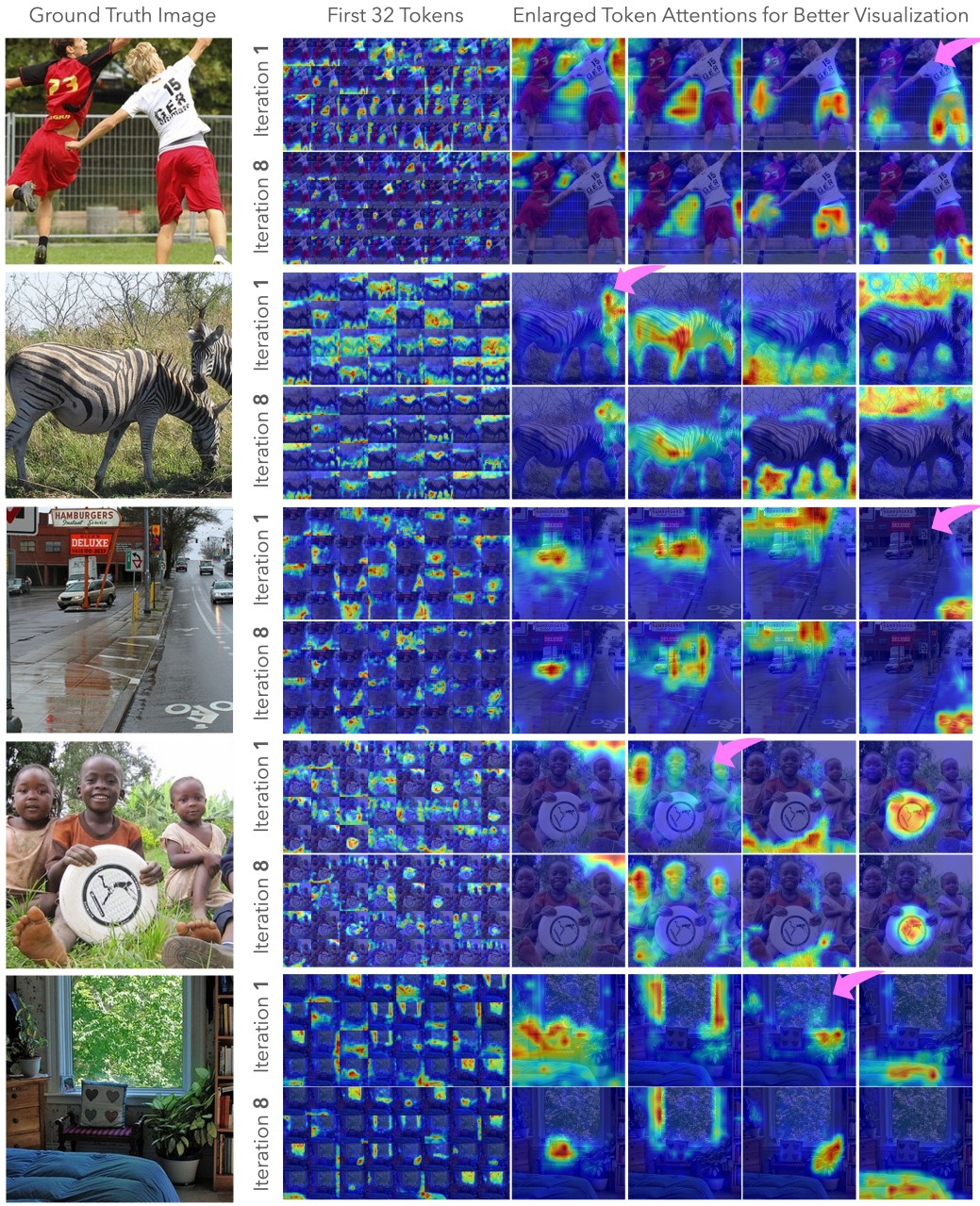

Figure 15: **Recurrent processing of Latent Tokens on COCO Validation Set:** Recurrent processing with increasing computational tokens leads to specialization of attention. We showcase all the first 32 tokens after the $1^{st}$ & $8^{th}$ iterations (out of total 256 tokens at the $8^{th}$ iteration). (Left) A broader comparison b/w iteration 1 vs 8 attention maps highlight emerging sparsity in attention.

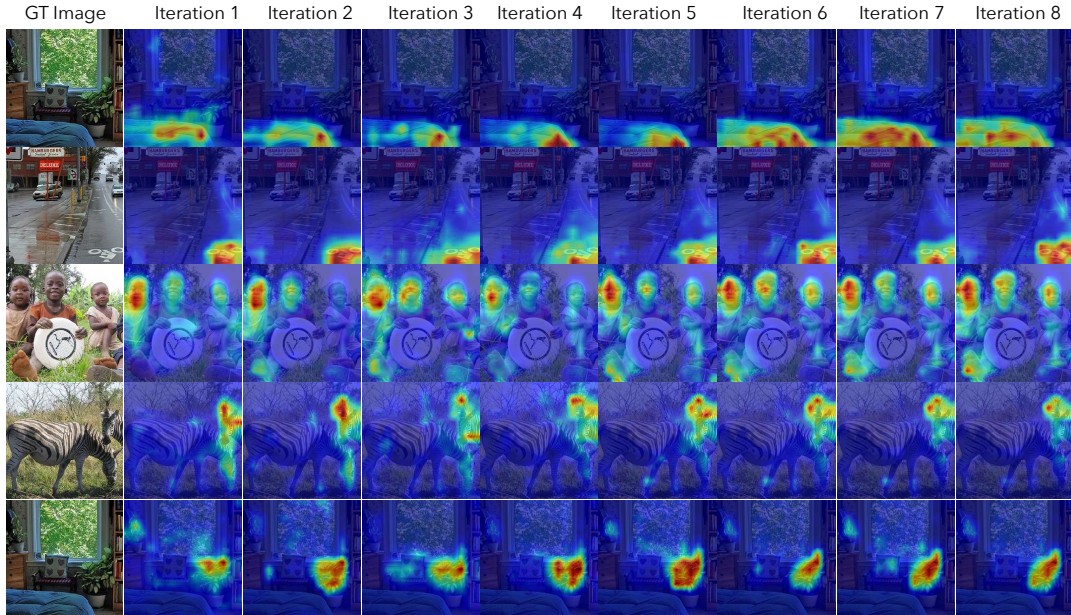

Figure 16: **Token Visualization over Multiple Recurrent Iteration:** Recurrent processing leads to sparse attention & specialization of tokens to localised objects, parts etc. For example – recurrence led to near-perfect bed segmentation (row 1), bicycle-sign segmentation (row 2), improved human segmentation (row 3), zebra-instance segmentation (row 4), and potted-plants segmentation (row 5).

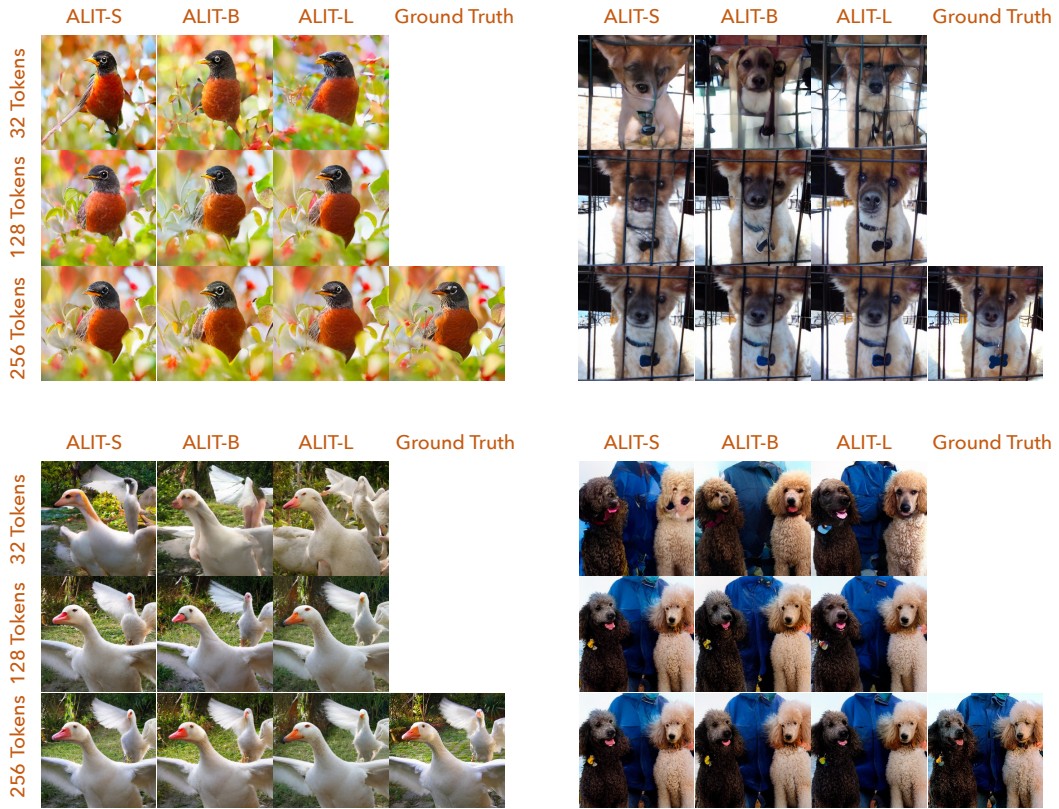

Figure 17: **Effect of Model Size on Image Reconstruction Quality.** Increasing model size enhances reconstruction quality. This visualization uses VQGAN as the base tokenizer (discrete 2D) distilled into discrete 1D latent tokens. With 256 tokens, the reconstruction quality closely aligns with VQGAN results and approaches ground truth, even for complex images, supporting the use of adaptive representations. (Zoom in for details.)

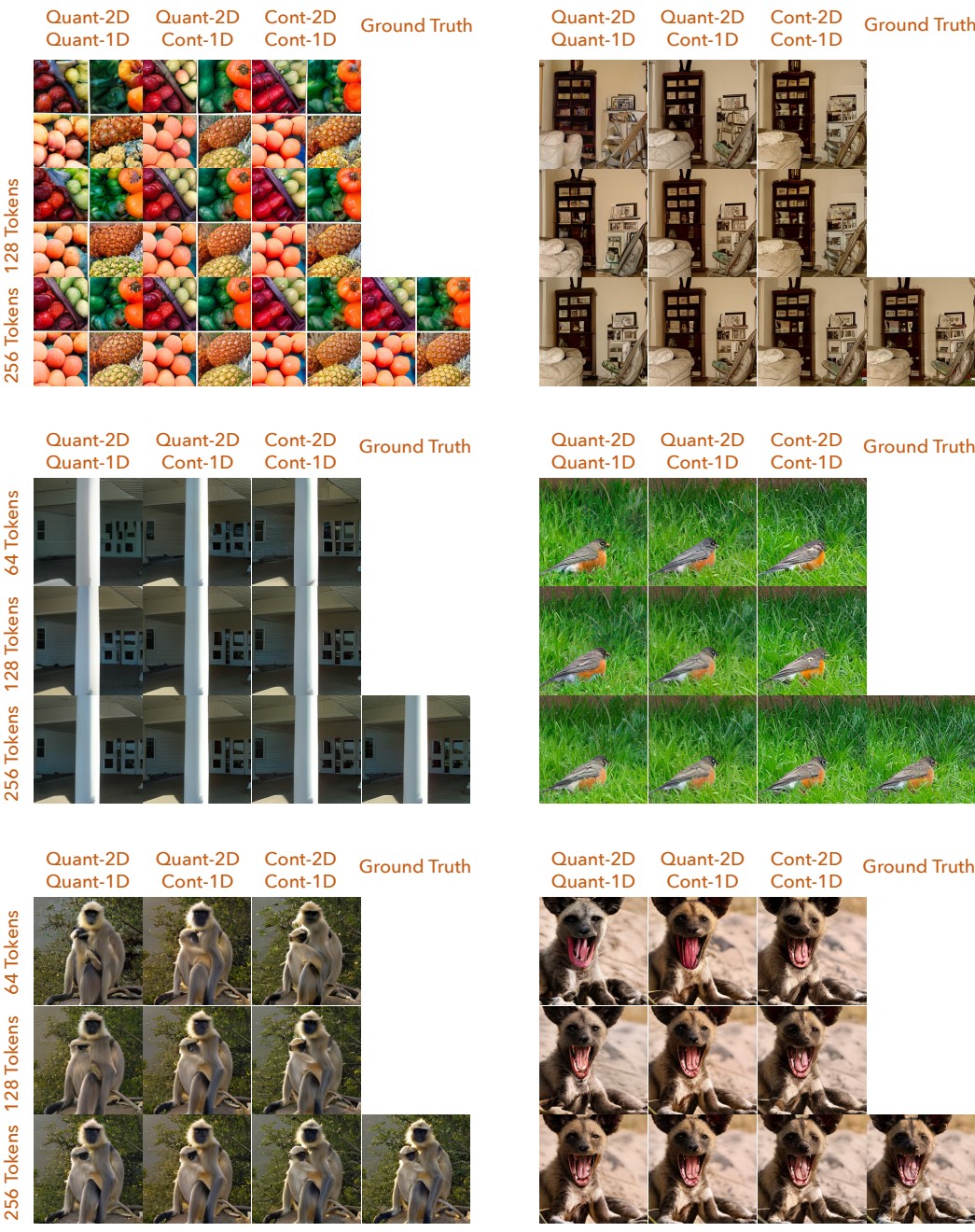

Figure 18: **Impact of Discrete vs. Continuous Tokenization on Image Reconstruction.** Continuous 1D latent representations yield the best reconstructions overall. The advantage of using VAE as the base tokenizer (third column of Cont-2D + Cont-1D) becomes more significant at higher token counts. VQGAN (discrete 2D) combined with continuous 1D (second column) also performs well, especially at lower token counts, possibly due to the learning ease of the discrete cross-entropy training framework. Note: ALIT-S is used in this visualization, and reconstructions may improve with a larger model. (Zoom in for details.)

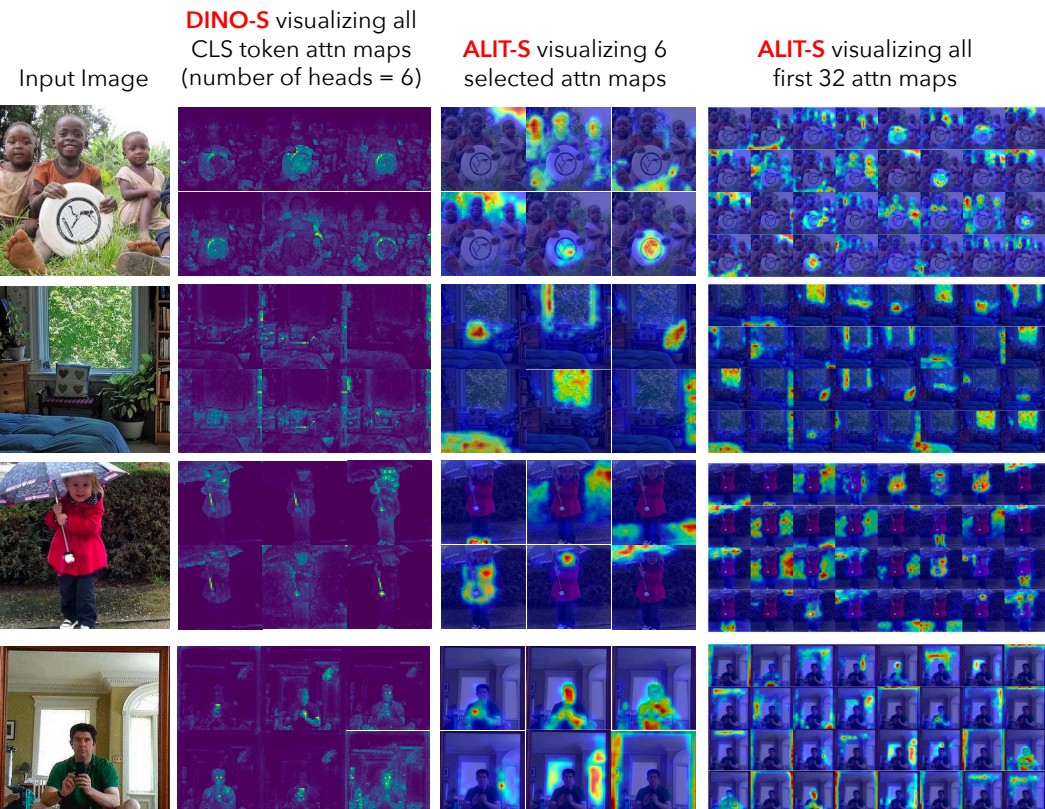

Figure 19: **DINO vs ALIT Token Visualization:** Compared to DINO-S, which has one class token and 6 attention-heads per token, leading to 6 token-attention maps, we have arbitrary attention maps (= number of latent tokens). *Dino being trained with self-distillation implicitly has some bias towards attending patches which lead to better classification, like "attending to phone and man for implicit selfie representation", while ALIT being trained with reconstruction even attends to the small switch on the wall in the background behind the man.* **Note:** we are not an alternative to DINO, both approaches have orthogonal contributions.

| Input Image (from Internet) | 64 Token Reconstruction | 128 Token Reconstruction | 256 Token Reconstruction |

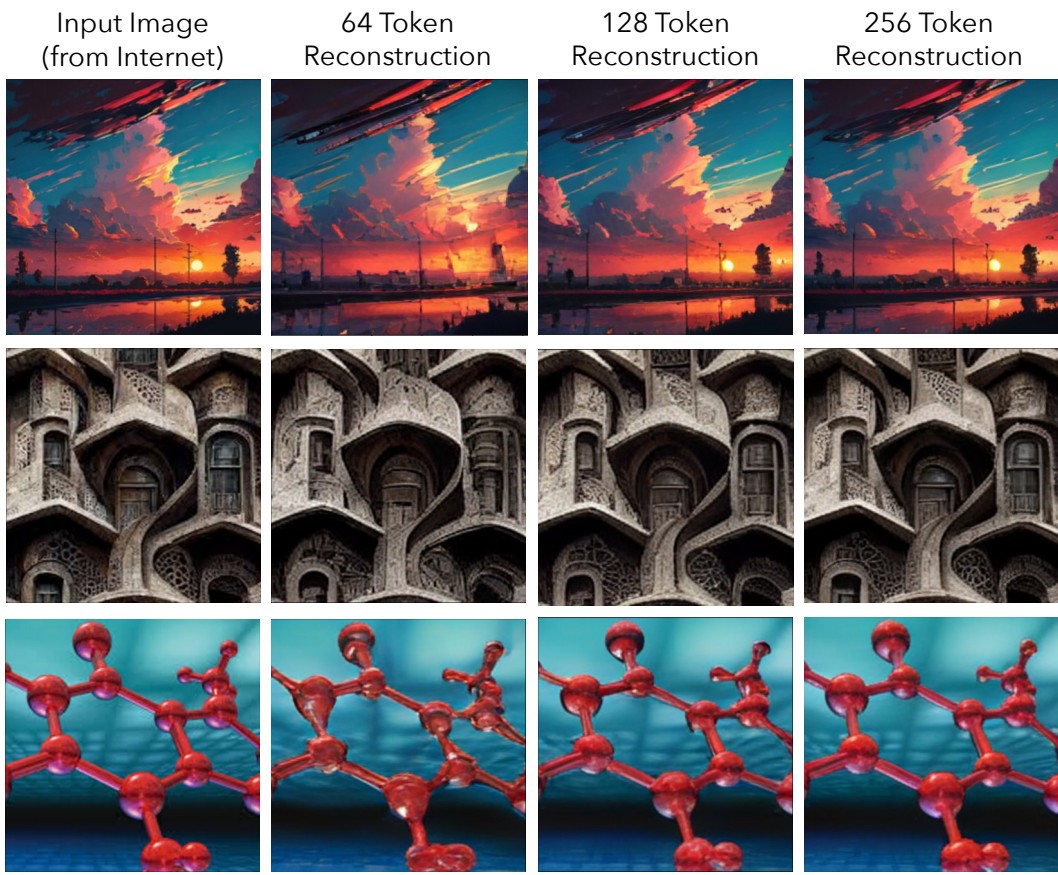

Figure 20: **Generalization of ALIT trained on Imagenet100 on Internet Images:** ALIT performs quite well even on OOD imagesmost times, with great potential of being an "any-image" generalizable adaptive tokenizer with some scaling.

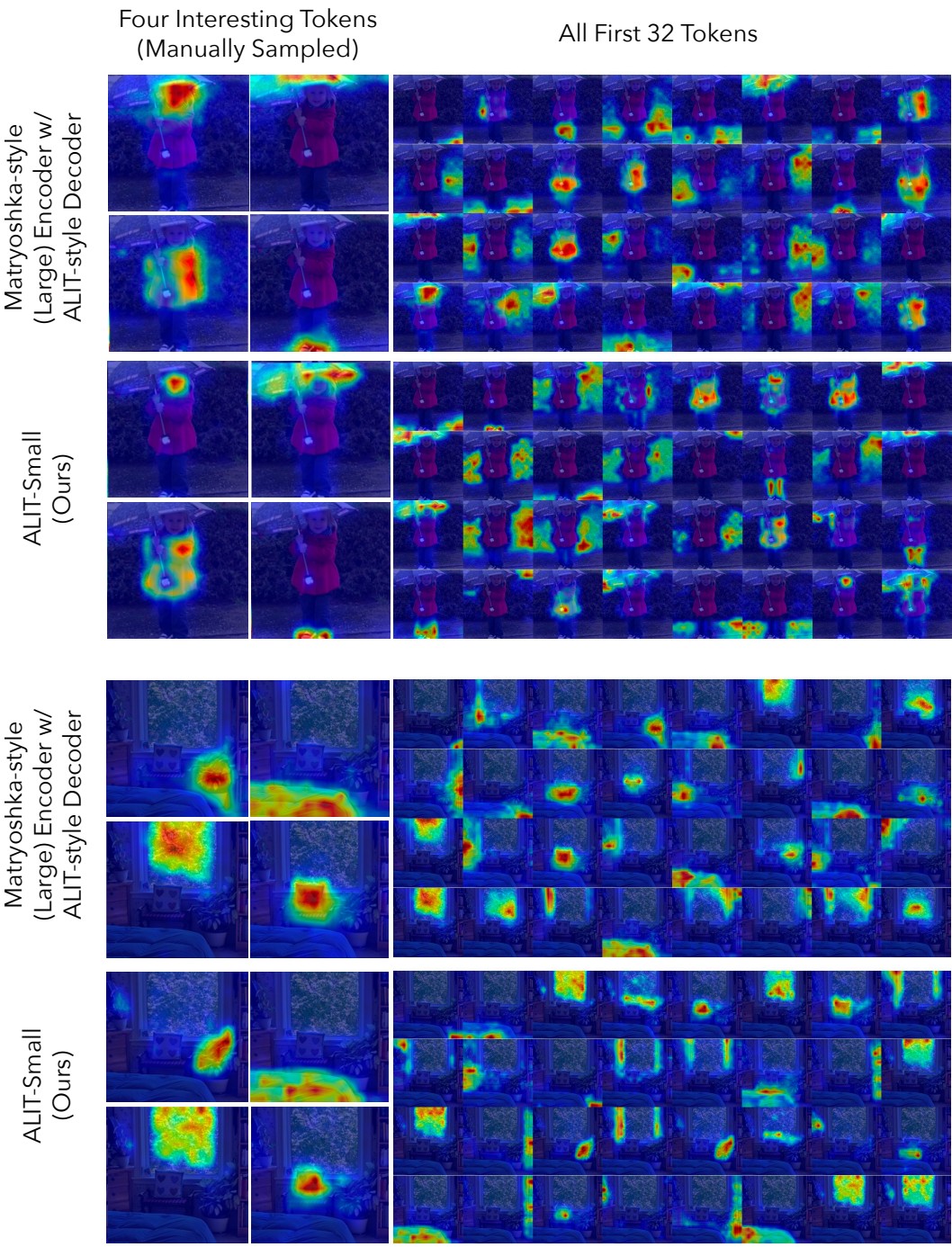

Figure 21: **Comparing ALIT adaptive tokenizer vs Matryoshka-Style tokenizer:** Thanks to recurrent processing in presence of additonal compute (a.k.a tokens), ALIT learns more sharper attention maps over multiple iterations. Unlike ALIT where the first 32 tokens imporve over iterations, Matryoshka-style approach learn a fixed-length (static) embedding in one go. **Note: This is ALIT-style decoding with Matryoshka-style encoding.** One could also combine Matryoshka-Style encoding with recurrent encoding, which would be interesting future work.

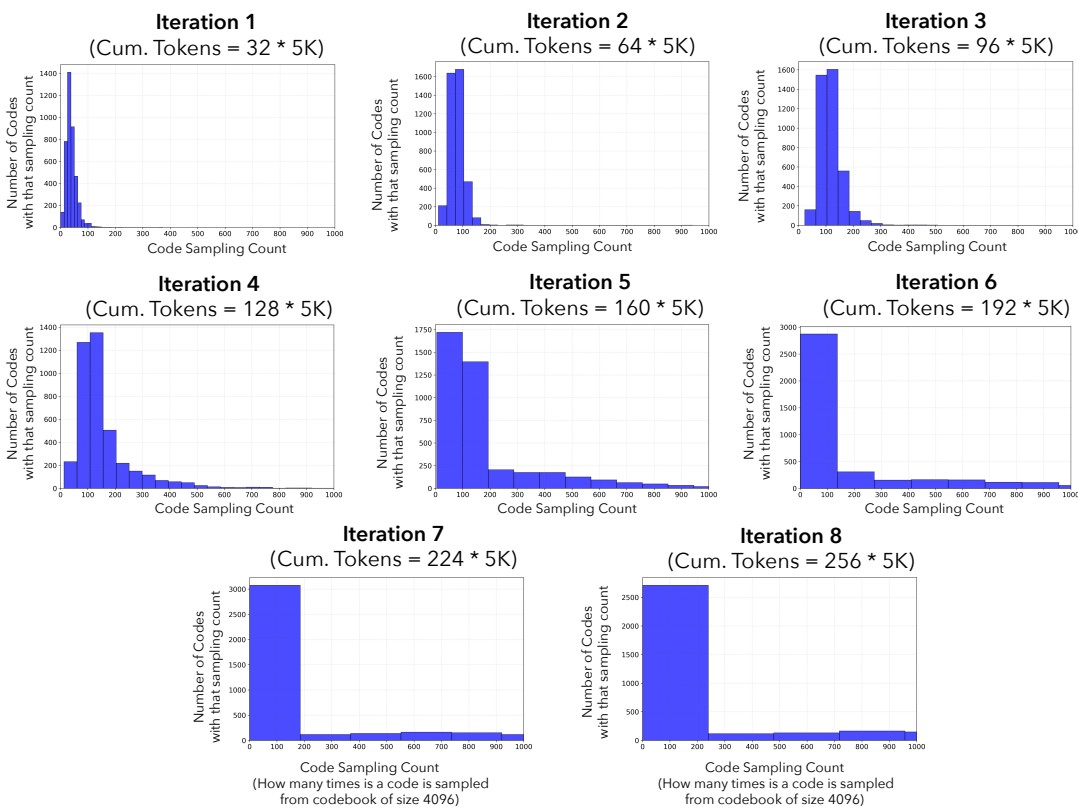

Figure 22: **Codebook Utilization at different recurrent iterations by ALIT-Small:** As mentioned before, recurrent processing with adaptive memory (tokens), leads to token specialization. This can also be seen from the fact — at larger iterations, there are more number of higher frequency tokens. **When you have more memory / tokens, some of the memory can be utilized for specialization.** Note – almost no code is sampled 0 times, that's just a artifact of low x-axis tick sampling. "*5K" = cumulative tokens from 5K Imagenet100 validation images.

