# OpenReview forum: "Adaptive Length Image Tokenization via Recurrent Allocation"
_ICLR.cc/2025/Conference — ICLR 2025 Poster_

### Official Review · Reviewer_6BgE · 2024-10-30

**Soundness:** 4
**Presentation:** 3
**Contribution:** 3
**Rating:** 6
**Confidence:** 3

**Summary:**

This paper discusses a new approach to variable-length visual token representation learning that addresses the limitations of fixed-length representations for images. The method, AVT, processes image tokens recursively, refining them into 1D latent tokens over multiple iterations while adaptively increasing capacity by adding new tokens. The iterative process potentially facilitates object and part discovery based on image entropy and context.

**Strengths:**

1. The paper proposed a novel method, AVT, which distills 2D image tokens into 1D variable-length latent tokens, an important topic for Visual Representation Learning.
2. The experiments are solid and comprehensive.
3. The writing is clear and concise.

**Weaknesses:**

1. The reconstruction effect, FID, of their method AVT in Table 1 has no obvious outstanding improvement compared with existing methods, VQGAN and Tiktok, under the same number of tokens.
2. Experimental results, in section 5 Further Experiments & Ablations, can be displayed in the form of figures and tables, which will make them more intuitive.

**Questions:**

1. How to determine how many iterations are needed for each picture? I see that you have done a lot of interesting experiments, but currently, it can only be set manually through experiments. Can it be selected adaptively based on the amount of information, for example, information entropy for the image?
2. Compared to 2D fixed-length tokenizers like VQGAN, or distilled 1D fixed-length tokenization methods like Perciver. How much calculation and storage space does the AVT method increase? Is it much slower than the above two methods through iterative reconstruction in AVT?
3. Tab.1 shows that the FID reconstructed through your method is worse than Tiktok and VQGAN. As you explain in line 453, the datasets for training and evaluation are different. Have you tried the experiment under the same situation?

---

> ### Author Response · Authors · 2024-11-21
> **Response to all the reviewer queries (1/N)**
>
> We thank the reviewer for appreciating the novelty of the proposed adaptive tokenizer, and for acknowledging the comprehensive set of experiments backing the proposed approach. We now try to answer all your queries:
>
> > The reconstruction effect, FID, of their method AVT in Table 1 has no obvious outstanding improvement compared with existing methods, VQ-GAN and Titok, under the same number of tokens.
>
> \
> Firstly, as acknowledged by the reviewers, **the primary goal of the paper is not to improve the FID of existing image tokenizers** but to introduce a novel adaptive and recurrent image tokenization approach that assigns varying numbers of tokens or representation capacities to different images. As noted by all reviewers, our adaptive tokenizer, despite its dynamic token-length predictions (with a single model for all variable-sized predictions), **performs comparably to fixed-length tokenizers** like VQ-GAN and TikTok on the FID metric.
>
> The two main reasons for not seeing major improvements, or slightly lower reconstruction metrics like L1 loss and FID, are:
> (a) Our tokenizer’s encoder/decoder layers are amortized (same layers are used across different RNN iterations) to predict a variable number of tokens. The same is true for the recently implemented Matryoshka baseline.
> (b) Our tokenizer is trained on the smaller Imagenet100 dataset compared to others.
>
> We believe that scaling could close any small gap between fixed-length and adaptive-length tokenizers, and **any improvement made to fixed-length tokenizers could be transferred to adaptive-length tokenizers (for example, by distilling from a better model)**.
>
> **Scaling Observations:** Figure 9 shows some scaling laws, demonstrating that training the model on larger datasets for longer durations and using continuous tokenization yields substantial improvements in FID.
> For instance, in Figure 9 (Plot 3), for 32-token reconstructions, FID improves from **53.x (stage-1 on Imagenet-100) --> 30.x (stage-1 on full ImageNet-1K)**, significantly closing the gap with baseline methods (trained on full ImageNet). The second stage also results in a significant drop in FID, from **53.x (stage-1 only trained on Imagenet-100) --> 22.x (stage-1 + stage-2 trained on Imagenet-100)**. We apologize if this wasn't clear from the scaling plots.
>
> Another example: **VQGAN with continuous 1D tokens** results at 128 tokens (shown in Table 4, Row 4) already outperform **Titok at 128 tokens**.
>
> ***
> > Experimental results, in section 5 Further Experiments & Ablations, can be displayed in the form of figures and tables, which will make them more intuitive.
>
> Thanks for the great suggestion, we completely agree! We've updated the supplementary materials to include more visual examples and tables for various subsections of Section 5. Moreover, Table 1 (for reconstruction metric comparison), Figures 7, 8, 12, and 15 (for token visualization) and Figure 9 for scaling laws, ablating continuous vs adaptive tokens –– are associated with Section 5.
>
> ***
> > How to determine how many iterations are needed for each picture? I see that you have done a lot of interesting experiments, but currently, it can only be set manually through experiments. Can it be selected adaptively based on the amount of information, for example, information entropy for the image?
>
> \
> Thanks for the great suggestion and for bringing this up!
>
> **Automatic token selection per-image at Inference**: We already presented an automatic token selection criterion in the paper (via Figure 5) but, on reflection, didn’t explicitly highlight it as “automatic.” The criterion is based on a reconstruction loss threshold: we decode the image using varying token counts and select the minimum token length that satisfies the objective of reconstruction loss < threshold. This approach ensures an efficient allocation of tokens while meeting quality requirements. Additionally, we will provide this automatic token selection as a user-friendly API in the released code.
> **Figures 3, 10, and 11 demonstrate that reconstruction loss correlates strongly with image entropy**, making token selection by reconstruction loss effectively equivalent to selection based on image entropy. Furthermore, we tested our adaptive tokenizer on internet images (Figure 20)—significantly different from the ImageNet100 dataset used for training—and observed that the automatically selected (variable) token counts effectively represent these diverse input images.
> ***

---

> ### Author Response · Authors · 2024-11-21
> **Response to all the reviewer queries & Conclusion (2/N)**
>
> > Compared to 2D fixed-length tokenizers like VQGAN, or distilled 1D fixed-length tokenization methods like Perciver. How much calculation and storage space does the AVT method increase? Is it much slower than the above two methods through iterative reconstruction in AVT?
>
> \
> \
> Thanks for the great question!\
> The overhead of running an additional iteration of the adaptive tokenizer encoder is actually quite small: **4ms run time and <+30 gflops on a single h100 gpu**. Here are the comparisons with a fixed length baseline:
>
> \
> Inference time comparison (in milli-seconds, on single h100 gpu, fp32 precision) for single image encoding:
> Algorithm | Iteration 1 | Iteration 2 | Iteration 3 | Iteration 4 | Iteration 5 | Iteration 6 | Iteration 7 | Iteration 8
> |----------|----------|----------|----------|----------|----------|----------|----------|----------|
> Titok-l-32| 9.54
> Ours-S| 6.84|10.40|14.13|18.04|22.03|25.99|30.20|34.74
>
>
> \
> GFlops Comparison (on h100 gpu) for single image encoding:
> Algorithm | Iteration 1 | Iteration 2 | Iteration 3 | Iteration 4 | Iteration 5 | Iteration 6 | Iteration 7 | Iteration 8
> |----------|----------|----------|----------|----------|----------|----------|----------|----------|
> Titok-l-32| 58.44
> Ours-S| 83.88|105.45|129.16|155.03|183.04|213.21|245.53|278.00
>
> \
> **Some points regarding the tables:**
> * For 32 tokens, we our latent-distillation encoder has 8 transformer layers much smaller than Titok-L-32 (with same number of attention heads, feature dim and same number of tokens). Despite that, we have more GFlops because of using  VQGAN image encoder + a small additional patch embedding layer as the image 2D embedding. We can switch to Titok-L-32 style image encoding and reduce upto 50 gflops.
> * Overall, +30 gflops overhead per iteration is very less for modern compute and can be further reduced by using techniques like KV caching previous layer attentions etc.
> * The reported metrics for inference time are using fp32 precision. With lower precision, inference time can be further lowered without loss in accuracy.
> ***
>
> >Tab.1 shows that the FID reconstructed through your method is worse than Tiktok and VQGAN. As you explain in line 453, the datasets for training and evaluation are different. Have you tried the experiment under the same situation?
>
> We did try running Titok-B-64 model on Imagenet100. Compared to the model on full Imagenet-1K (with FID 8.22), the FID dropped to 9.82, showcasing that training on larger dataset helps!
>
> \
> **Overall, we are grateful to the reviewer for their strong positive comments and constructive feedback. We hope we have addressed all your queries. Please let us know if there’s anything else we can do to assist in increasing the rating, more in-alignment with the strengths section you provided.**
>
> To conclude, we would also like to point the reviewers to some interesting new figures as mentioned in the general response (Figure 16 onwards in supplementary)
>
> Best regards!

---

> > ### Author Response · Authors · 2024-11-23
> > **Response to Reviewer 6BgE**
> >
> > We really appreciate your strong and detailed positive feedback, as well as your valuable feedback.
> > We hope we have answered all your queries. With the discussion period ending soon, please let us know if there is anything further we can do to help refine your rating to better reflect the strengths you highlighted.
> >
> > Best Regards!

---

> > > ### Comment · Reviewer_6BgE · 2024-11-25
> > >
> > > Thank you for addressing all my questions and clarifying my doubts about your paper. I decide to maintain my previous rating.

---

> > > > ### Author Response · Authors · 2024-11-26
> > > > **Response to Reviewer 6BgE**
> > > >
> > > > We are grateful to the reviewer for their response and for acknowledging that the paper is novel, experiments are solid & comprehensive, writing is clear & concise, and the rebuttal resolves all questions.
> > > >
> > > > We would greatly appreciate any additional feedback on how we can further improve the paper and would kindly like to understand why it is still considered a "borderline accept".
> > > >
> > > > Thank you once again for your time and thoughtful review.
> > > >
> > > > Best Regards!

---

### Official Review · Reviewer_46Y3 · 2024-11-02

**Soundness:** 3
**Presentation:** 3
**Contribution:** 3
**Rating:** 6
**Confidence:** 3

**Summary:**

The paper explores the concept of adaptive and recursive representations for image processing, focusing on how many tokens are necessary to effectively represent an image. The authors conduct a series of experiments to evaluate their approach, starting with image reconstruction tasks and linear probing for Top-1 classification. They investigate the impact of using continuous versus discrete tokenizers on reconstruction tasks and analyze the learned tokens' alignment with ground truth segmentation, suggesting potential for object discovery. The role of recurrent processing in achieving these results is also examined.

A key contribution of the paper is the demonstration of a correlation between an image's perceptual complexity and its compressibility, as theorized by Schmidhuber's Low Complexity Art theory. The authors show that images with lower complexity can be reconstructed with fewer tokens, while more complex images require a higher number of tokens for accurate reconstruction. This is evidenced by experiments using out-of-distribution images from the PeopleART dataset, where the model trained on ImageNet-100 successfully reconstructs images of varying complexities with different token capacities.

The paper highlights two main observations: (a) as image complexity increases, using fewer tokens results in higher reconstruction errors, necessitating a larger computational budget; and (b) for a fixed image complexity, increasing the number of tokens reduces the reconstruction loss, showcasing the efficiency of the adaptive representation model. These findings underscore the potential of the proposed approach in efficiently handling images of varying complexities by adjusting the number of tokens used for representation.

**Strengths:**

Originality: The paper introduces an innovative method for image representation by employing a multi-stage training procedure that combines VQGAN and distillation techniques. The approach of using adaptive and recursive representations, along with the introduction of a shared 1D codebook for token quantization, is a novel contribution to the field. The concept of dynamically halting token processing based on reconstruction quality adds a unique aspect to the methodology, allowing for more efficient and tailored image representation.

Quality: The quality of the research is demonstrated through comprehensive experiments and ablation studies. The paper provides a thorough evaluation of the proposed method, including image reconstruction experiments, linear probing for classification, and analysis of learned tokens for object discovery. The results, such as the high mean IOU achieved in attention map alignment with ImageNet-S GT segmentation, underscore the effectiveness of the approach. The use of GAN loss to enhance photo-realism further attests to the robustness of the training procedure.

Clarity: The paper is well-structured and clearly articulates the training procedure and experimental setup. The inclusion of figures, such as the adaptive visual tokenizer diagram, aids in understanding the complex process of 2D to 1D to 2D distillation. However, some sections could benefit from additional explanations, particularly for readers unfamiliar with the underlying concepts of VQGAN and distillation techniques. The appendix is referenced for implementation details, which is helpful for those seeking to replicate the study.

Significance: The significance of the paper lies in its potential impact on the field of image processing and representation learning. By demonstrating that adaptive representations can lead to emergent properties like object discovery, the research opens new avenues for exploring how different tokenization strategies can enhance image understanding. The ability to achieve high alignment with segmentation tasks without explicit optimization for segmentation highlights the broader applicability of the method in various computer vision tasks.

**Weaknesses:**

Limited Evaluation Metrics: The evaluation of the model is primarily focused on image reconstruction and classification tasks. While these are important, the paper could benefit from a broader range of evaluation metrics, particularly those that assess the quality and utility of the learned representations in more diverse and complex tasks.

Tokenization Approach: The paper discusses the use of continuous vs. discrete tokenizers but does not provide a comprehensive analysis of the advantages and disadvantages of each approach. A more detailed comparison could help in understanding the trade-offs involved and the scenarios where one might be preferred over the other.

Sensitivity to Model Strength: The findings suggest that stronger models are more sensitive to fewer-token reconstructions, which could be a limitation in practical applications where computational resources are constrained. The paper could explore strategies to mitigate this sensitivity and improve the robustness of the approach across different model strengths.

Generalization to Other Tasks: The paper notes that classification tasks do not align well with dense tasks like depth estimation and reconstruction. This indicates a potential limitation in the generalization of the learned representations to various tasks. Further exploration into how these representations can be adapted or extended to perform well across a wider range of tasks would be beneficial.

**Questions:**

Please refer to the weakness.

---

> ### Author Response · Authors · 2024-11-21
> **Response to all reviewer queries (1/N)**
>
> We thank the reviewer for their strong positive feedback overall. We now try to answer all the queries:
>
> > Limited Evaluation Metrics: The evaluation of the model is primarily focused on image reconstruction and classification tasks. While these are important, the paper could benefit from a broader range of evaluation metrics, particularly those that assess the quality and utility of the learned representations in more diverse and complex tasks.
>
> \
> We already explored several tasks such as **Classification**, **Reconstruction**, **Depth Estimation**, and **VLM** by first passing the learned tokens through the reconstruction decoder network, followed by training the network for various downstream tasks.
>
> That said, we acknowledge that jointly training both networks for different downstream tasks could lead to better token compression and more efficient token sampling for each specific task (as highlighted in the new Figure 16). In fact, we are very interested in further investigating adaptive tokenizers as a **universal tokenizer** and jointly training for multiple tasks. This has also garnered interest from general audience in the field, as a potential next work. However, to keep the current paper focused and crisp, we have decided to leave this exploration for future work.
> ***
> > Tokenization Approach: The paper discusses the use of continuous vs. discrete tokenizers but does not provide a comprehensive analysis of the advantages and disadvantages of each approach. A more detailed comparison could help in understanding the trade-offs involved and the scenarios where one might be preferred over the other.
>
> \
> Thanks for the great question!
> We showcased a reconstruction-metric-based comparison between continuous and discrete tokenization at both the 2D token level and the 1D token level in **Figure 9 (plot 4)** and as a table in the supplementary material. Additionally, we have now added a **new qualitative visualization comparing discrete vs. continuous tokenization in the supplementary (Figure 18).**
>
> Here are some key observations regarding them:
>
> -   **Continuous tokens at the 1D level** (regardless of whether the tokenization is continuous or quantized at the 2D level) consistently lead to the best FID performance.
> -   **Continuous tokens tend to perform better for reconstruction** and generation tasks, preserving the maximum amount of detail. The trade-off, however, is that they do not offer any compression.
> -   In contrast, **quantization results in a decent compression rate**, but there is a slight loss of details. Quantization, originally introduced in signal processing and communication, is important for **error-bounding theoretically.
>
> Investigating the utility of quantization and learned codebooks for tasks like visual abstract reasoning and understanding is a promising area for future work. Furthermore, an interesting insight we shared in Section 4 is that the learned quantized codebook helps us identify **familiar vs. less familiar** samples during inference, which could be a valuable feature for certain tasks.
> ***
> > Sensitivity to Model Strength: The findings suggest that stronger models are more sensitive to fewer-token reconstructions, which could be a limitation in practical applications where computational resources are constrained. The paper could explore strategies to mitigate this sensitivity and improve the robustness of the approach across different model strengths.
>
> \
> Thanks for the great suggestion!
> We rather see this as a strength of our paper, as it **opens the door for future works aimed at making models more robust to variable token compression of the input data.**
>
> Just to clarify: stronger models inherently require representations that preserve maximum detail of the input, as they are more knowledgeable in their task. However, if a model is weak in a certain aspect—say, for instance, it cannot detect a particular object—then, even if the tokenizer learns to ignore that object, the performance of the weak model would not be significantly affected. This is because the weak model might not rely on that specific feature to make predictions. Thus, in Section 4, we highlighted that the required representation depends not only on the tokenizer but also on the strength of the downstream model.
> ***

---

> ### Author Response · Authors · 2024-11-21
> **Response to all reviewer questions & Conclusion (2/N)**
>
> > Generalization to Other Tasks: The paper notes that classification tasks do not align well with dense tasks like depth estimation and reconstruction. This indicates a potential limitation in the generalization of the learned representations to various tasks. Further exploration into how these representations can be adapted or extended to perform well across a wider range of tasks would be beneficial.
>
> \
> We apologize for slightly incorrect writing of that supplementary paragraph leading to incorrect insight (Fig. 13 and Fig. 14 were correct). Thanks for pointing it out. We have updated the supplementary with the correct paragraph (copied from updated supplementary for ease of review):
>
> \
> **Representational capacity depends on downstream tasks:** In addition to Fig. 6 in the main paper, where we evaluated Reconstruction Loss Thresholding as a Token Selection Criterion (TSC), we further assess GT-oracle-based Classification and pseudo-GT-oracle-based Depth Error metrics as TSC through Fig. 14 and Fig. 15. For Top-X Classification as TSC, an image receives 32 latent tokens if ResNet-18’s top-X predictions from the 32-token reconstruction include the ground truth class. **Key Takeaways** – Compared to depth, reconstruction loss, and FID as the tasks of interest (TOI), the decrease in classification accuracy with a reduction in dataset representation capacity (cumulative tokens per dataset) is relatively smaller due to classification being a coarser task. **When TSC  != TOI, using different TSCs to sample the same fraction of tokens yields similar TOI performance. For example, FID and Reconstruction Loss at 40% of max token capacity are comparable (compare 2nd , 3rd plots of Fig. 14 and Fig. 15), regardless of whether depth or classification was the token-selection criterion. However, compared to Top-X classification, choosing the more granular tasks of depth and reconstruction as TSC provide finer support for token sampling – therefore better/ diverse performance range for the TOI. Reconstruction loss serves as self-supervised TSC.**
>
> We hope the **bolded text** fixes the confusion. And just to clarify, we do not explicitly train for classification task.
>
> \
> \
> **Overall, we are grateful to the reviewer for their strong positive comments and constructive feedback. We hope we have addressed all your queries. Please let us know if there’s anything else we can do to assist in increasing the rating, more in-alignment with the detailed strengths section you provided.**.
>
> To conclude, we would also like to point the reviewers to some interesting new figures as mentioned in the general response (Figure 16 onwards in supplementary)
>
> Best regards!

---

> > ### Comment · Reviewer_46Y3 · 2024-11-21
> >
> > The author's response resolved my issue, and I have decided to maintain my rating.

---

> > > ### Author Response · Authors · 2024-11-21
> > > **Response to Reviewer 46Y3**
> > >
> > > We thank the reviewer for their prompt response and for accepting our rebuttal, addressing their queries.
> > >
> > > Please let us know if there is anything further we can do to help increase the rating, more in alignment with the strengths you mentioned.
> > >
> > > Best Regards!

---

> > > > ### Author Response · Authors · 2024-11-26
> > > > **Response to Reviewer 46Y3**
> > > >
> > > > We are grateful to the reviewer for their response and for acknowledging the paper's originality, quality, clarity, significance, and that the rebuttal resolves all questions.
> > > >
> > > > We would greatly appreciate any additional feedback on how we can further improve the paper and would kindly like to understand why it is still considered a "borderline accept."
> > > >
> > > > Thank you once again for your time and thoughtful review.
> > > >
> > > > Best regards!

---

### Official Review · Reviewer_tXpo · 2024-11-04

**Soundness:** 2
**Presentation:** 2
**Contribution:** 3
**Rating:** 5
**Confidence:** 3

**Summary:**

Representation learning models assign fixed-length representations to samples regardless of their information content. In contrast, the authors aim to present a method for learning variable-length tokenized representations for 2D images.

The paper introduces the Adaptive Visual Tokenizer (AVT), which recursively distills 2D images into 1D latent tokens through an encoder-decoder model trained using a reconstruction objective. To achieve variable-length tokenized representations, the authors add new tokens at every iteration of the recurrence. The authors allow optional masking of specific 2D tokens that are well-reconstructed, enabling "dynamic halting" per token.

Through experiments with their method, the authors provide several insights relating to the adaptive representation capacity of AVT:
- More complex images require more tokens to reconstruct.
- OOD images require more tokens than in-distribution ones to reconstruct.
- As the downstream criterion for accepting the representation increases in complexity, more tokens are required.
- Optimal adaptation for performance occurs when the token selection criterion is relevant to the task of interest.
- Weaker models exhibit smaller relative performance drops with reduced token counts than stronger ones.

The authors also compare AVT reconstructions and linear probe performance with various baselines. They analyze the impact of choosing continuous vs discrete tokenizers. Analyzing attention maps of 1D latent tokens to 2D image tokens reveals that tokens attend to distinct semantically meaningful objects or parts, suggesting the emergence of object discovery. Finally, the authors show that broad attention of early tokens specializes to focus on sparser, more meaningful regions, when new tokens are introduced as the recurrence proceeds.

**Strengths:**

- The paper presents thorough analysis of the adaptive representation capacity of the proposed method.
- The authors show that their method can lead to the emergence of tokens that exhibit useful semantics.

**Weaknesses:**

Baselines are very limited.
- The paper is motivated from a representation learning perspective, but there is no comparison with representation learning methods beyond two families of generative models. For vision, methods like CLIP [1], MoCo [2], DINO [3], and MAE [4] (which is also reconstruction-based) should be compared for downstream tasks like linear probe.
- There is no comparison with other variable-length representation baselines, such as with Matryoshka representation learning [5], despite noting the relevance in the paper.

The comparisons with current baselines have very mixed results.
- In Table 1, all baselines perform better than the authors for a specific token count, and many remain better even when compared to the max token count of AVT. The authors note that longer training on larger datasets with deeper networks could bridge the gap. Still, it is not apparent if that would be enough to outperform the baselines, especially ensuring that the training budgets remain fair.
- Linear probe performance in Section 5 is at par with baselines, but often requiring more tokens and recurrent processing.

Discussion of gradients
- There is no discussion of how gradients are treated within the recurrent processing. It is unclear if the authors backpropagate gradients from future iterations all the way back or if they stop the gradient flow between each iteration.

The insight about weak and strong models is not very convincing.
- Weaker models exhibiting smaller relative performance drops with reduced token counts could simply indicate that weak models, due to being weak, do not reach the same peak performances as the strong models. This would result in their performance with more tokens remaining similar to their initial performance.
- In this case, the statement should be that weaker models exhibit lower performance than stronger models, and they reach their performance cap sooner than stronger models as the number of tokens increases, which is not surprising.
- Looking at the absolute performance metrics can provide some insights here.

Dynamic halting is unclear.
- Figure 1 states that dynamic halting is optional, but a clear discussion of when it should be enabled or disabled is missing.
- It is unclear if it is enabled for all experiments or some subset.

Complexity of the method.
- The proposed method is not simple, including multiple stages like learning to reconstruct pre-trained VQGAN tokens, then joint encoder-decoder training, and then GAN loss at some later stage in training.

Minor (no effect on score):
- "Variable-length token representations" is confusing due to ambiguity: it can mean representations per token variable length or a variable number of tokens. Perhaps "variable-length tokenized representations" is a less ambiguous term.
- The footnote marked on Classification on line 397 points to a footnote printed three pages prior, which is confusing and not immediately apparent.
- Table 1 has elements not adequately labeled or described in the caption: what is the # superscript? The caption should also clarify that the numbers under the dataset names are the number of tokens and not some varying attribute of the datasets.
- Relevant paper you might want to cite that also learns tokenized representations (fixed-length) where tokens show the emergence of distinct concepts: [6].

References:
[1] Radford, Kim, Hallacy, Ramesh, Goh, Agarwal, Sastry, Askell, Mishkin, Clark, Krueger, Sutskever.
Learning Transferable Visual Models from Natural Language Supervision. ICML 2021.
[2] Chen, Fan, Girshick, He. Improved Baselines with Momentum Contrastive Learning. CoRR 2020.
[3] Caron, Touvron, Misra, Jégou, Mairal, Bojanowski, Joulin. Emerging Properties in Self-Supervised Vision Transformers. ICCV 2021.
[4] He, Chen, Xie, Li, Dollár, Girshick. Masked Autoencoders are Scalable Vision Learners. CVPR 2022.
[5] Kusupati, Bhatt, Rege, Wallingford, Sinha, Ramanujan, Howard-Snyder, Chen, Kakade, Jain, Farhadi. Matryoshka Representation Learning. NeurIPS 2022.
[6] Vani, Nguyen, Lavoie, Krishna, Courville. SPARO: Selective Attention for Robust and Compositional Transformer Encodings for Vision. ECCV 2024.

**Questions:**

Currently, the number of new tokens added per iteration seems to be fixed to 32. What does the relationship between the number of tokens added per iteration and the total number of iterations, for a fixed max token count, look like? It would be interesting to see if trade-offs exist, like incurring higher computational costs by performing more iterations leading to significantly smaller "good enough" representations (each iteration adds a smaller amount of new tokens, but each token gets many more refinement updates).

---

> ### Author Response · Authors · 2024-11-21
> **Response to all reviewer questions (1/N)**
>
> We thank the reviewer for their valuable feedback. We have made a thorough attempt to address all the queries, including implementing the Matryoshka baseline, providing additional details on training and gradients (which got missed in the earlier submission due to oversight), additional ablations on dynamic halting etc.
>
> > The paper is motivated from a representation learning perspective, but there is no comparison with representation learning methods beyond two families of generative models. For vision, methods like CLIP [1], MoCo [2], DINO [3], and MAE [4] (which is also reconstruction-based) should be compared for downstream tasks like linear probe.
>
> \
> As acknowledged by all the reviewers, **we believe the core contribution of our work lies in the variable-length vision tokenizer.** Traditional representation learning approaches typically learn fixed-length representations, making them not direct baselines for our proposed method. However, we would like to highlight the following points regarding these existing works:
>
> -   First, our linear probing performance (60% on ImageNet) can be directly compared with the numbers reported in papers on DINO, MOCO, and MAE (which have higher linear probing performance as reported in their papers). These approaches focus on fixed-length representations and generally rely on discriminative methods (such as contrastive learning or self-distillation), or in the case of MAE, have been shown to have implicit local-contrastive behaviour due to the inpainting objective. In contrast, **our approach is fundamentally generative.**
> - To achieve the best classification performance, **our tokenizer learns a highly compressed representation** (32 tokens vs. per-patch 256 tokens), which intuitively requires a slightly larger decoding network. Additionally, since our tokens are not patch-bound, the classical technique of averaging all tokens before passing them through the linear classification layer might not be the most optimal. Despite this, we achieve solid linear probing results, on par with or slightly better than the Titok baseline.
> - That said, we would like to reiterate that the linear probing experiment is just a small part of the paper, and achieving the best linear probing accuracy is not the primary goal of our work. Similarly, the fact that MAE's linear probing accuracy is lower than prior works like DINO and MOCO does not diminish the core contributions of MAE.
> ***
> > Discussion of gradients: There is no discussion of how gradients are treated within the recurrent processing. It is unclear if the authors backpropagate gradients from future iterations all the way back or if they stop the gradient flow between each iteration.
>
> \
> We apologize for missing the training details in the supplementary at the time of submission and thank the reviewer for pointing this out. These details have now been added (in supplementary Section A.3), and we will release the entire codebase. We perform end-to-end backpropagation through all the rollout iterations. However, to save computational memory, we compute the reconstruction loss at one or a few rollout iterations, avoiding the forward and backward pass through the reconstruction decoder for all iterations during training.
> ***
>
> > The insight about weak and strong models is not very convincing.
>
>
> - We have already shared the absolute performance of each downstream model as the last point on the x-axis of the Figure 6 plots.
> -   We agree with the reviewer’s definition of weak and strong models—weak models exhibit poorer performance on ground truth (GT) data.
> -   We also align with the reviewer’s observation that weaker models perform poorly across all token counts. This validates that our adaptive tokenizer is trained effectively; fewer tokens do not suddenly outperform larger token counts -- adding more memory improves the representation.
> -   Finally, the key insight we presented in this section is that **the gradient of performance drop with token reduction is smaller for weaker models**. While this might not be immediately intuitive, it happens because potentially our learned adaptive tokens capture enough details for weaker model performance with fewer or earlier tokens, whereas the remaining details necessary for strong performance are encoded in later tokens (as memory increases).** We apologize if this wasn't clear from the paragraph at the first place and would fix the writing accordingly.**
> ***

---

> ### Author Response · Authors · 2024-11-21
> **Response to the reviewer queries (2/N)**
>
> > There is no comparison with other variable-length representation baselines, such as with Matryoshka representation learning [5], despite noting the relevance in the paper.
>
> We thank the reviewer for the thoughtful suggestion! We implemented the Matryoshka-style reconstruction baselines and demonstrate their performance below.
> * Firstly, we want to highlight the fact that Matryoshka-style approaches are originally proposed for classification task and have not been yet shown before for reconstruction –– implementing them for the task of reconstruction is not straight-forward (how to decode variable-size token representation to an image). We leverage our adaptive tokenizer style decoding ( i.e. using latent-distillation decoder) to decode variable Matryoshka tokens back to the image. In fact, application of **Matryoshka-style models** to reconstruction task is an ongoing submission to ICLR 2025 (and recently arXived) and the paper received no novelty comments.
> * Secondly, we also believe and are currently working on designing a new system which combines the Matryoshka style image encoding with our adaptive tokenizer. Therefore, our contributions are much different from that of Matryoshka -- we propose a recurrent tokenizer which can potentially be unrolled for infinite iterations at test time, while Matryoshka learns a fixed size embedding for the input and sample adaptive embeddings by masking different sub-embeddings.
> * We first share the FID metric comparison between ours and Matryoshka baseline on the in-distribution Imagenet100 dataset:
>
>
> Algorithm | 32 tokens | 64 tokens | 96 tokens | 128 tokens |160 tokens | 192 tokens | 224 tokens | 256 tokens
> |--------|---------|--------|-----------|-------|---------|--------|-----------|-------|
> Matryoska-S| 33.11|20.08|15.10|12.84|11.45|10.51|9.90|9.50
> Ours-S|**22.57**| **16.17**| **13.30**| **11.69**| **10.22**| **9.30**| **8.55**| **8.25**
> Matryoska-L| 22.85|14.51|11.63|**10.13**|**9.23**|8.83|8.42|8.11
> Ours-L|**19.70**| **13.92**| **11.39**| 10.41| **9.23**| **8.75**| **8.22**| **8.03**
>
>
> * The following tables shows FID comparison on out of distribution COCO and Wikipedia Image Text dataset. On OOD datasets, our small model performs much better than large Matryoshka model:
>
> \
> Trained on Imagenet100, Tested on COCO
> Algorithm | 64 tokens| 128 tokens| 256 tokens
> |--------|---------|--------|-----------|
> Matryoska-L| 23.05|14.43|10.66
> Ours-S|22.28| 14.22| 9.72
> Ours-L|**21.44**| **13.64**| **9.71**
>
> \
> Trained on Imagenet100, Tested on Wikipedia Image-Text (WIT)
> Algorithm | 64 tokens| 128 tokens| 256 tokens
> |--------|---------|--------|-----------|
> Matryoska-L| 62.02|48.21|40.28
> Ours-S|61.77| 47.91| **38.45**
> Ours-L|**60.11**| **47.52**| 38.69
>
> * Next, we would like to point the reviewer to Figure 21 for token-attention maps comparison between Matryoshka-style model and our tokenizer. Our adaptive tokenizer (ALIT) leads to sharper attentions, thanks to recurrent updates. As mentioned in the caption, the contribution of both approaches are different and they can be potentially combined together for a stringer adaptive tokenizer.
> * Finally, our recurrent tokenizer opens the door for infinite rollouts (longer test-time rollouts than train time) for highly complex images, which is an intriguing research question (Something, which Matryoshka-style models might not be able to address).
> ***
> > Linear probe performance in Section 5 is at par with baselines, but often requiring more tokens and recurrent processing.
>
> -   We believe this is a strength. Our model performs at par with fixed-length baselines, despite being amortized to predict a variable number of tokens per image. For example, our semiLarge models perform on par with the fixed-length Titok-L model at the same token count, even though **Titok-L is larger and throws more flops** for the 32-token representation compared to ours. We apologize if this wasn't clear, and will fix the writing accordingly.
>
> -   Secondly, the improvement in linear probing accuracy with more tokens is a desired behavior. **Representations should improve when provided with more memory**, and as expected, our 64 and 128 token representations achieve better performance than the 32-token representation.
> - Finally, as we mentioned before, because are representations are more compressed, we might need a longer decoder for the downstream classification task than just a Linear Layer. Moreover, our tokens are not patch-bound, so potentially simply averaging tokens is also not optimal. Investigating the best strategy for linear probing experiments for off-the-grid tokenization methods would be an interesting research direction.

---

> > ### Author Response · Authors · 2024-11-21
> > **Response to the reviewer queries (3/N)**
> >
> > > In Table 1, all baselines perform better than the authors for a specific token count, and many remain better even when compared to the max token count of AVT. The authors note that longer training on larger datasets with deeper networks could bridge the gap. Still, it is not apparent if that would be enough to outperform the baselines, especially ensuring that the training budgets remain fair.
> >
> > Firstly, as acknowledged by the reviewers, **the primary goal of the paper is not to improve the FID of existing image tokenizers** but to introduce a novel adaptive and recurrent image tokenization approach that assigns varying numbers of tokens or representation capacities to different images. As noted by all reviewers, our adaptive tokenizer, despite its dynamic token-length predictions (with a single model for all variable-sized predictions), **performs comparably to fixed-length tokenizers** like VQ-GAN and TikTok on the FID metric.
> >
> > The two main reasons for not seeing major improvements, or slightly lower reconstruction metrics like L1 loss and FID, are:
> > (a) Our tokenizer’s encoder/decoder layers are amortized (same layers are used across different RNN iterations) to predict a variable number of tokens. The same is true for the Matryoshka baseline.
> > (b) Our tokenizer is trained on the smaller Imagenet100 dataset compared to others.
> >
> > We believe that scaling could close any small gap between fixed-length and adaptive-length tokenizers, and **any improvement made to fixed-length tokenizers could be transferred to adaptive-length tokenizers (for example, by distilling from a better model)**.
> >
> > **Scaling Observations:** Figure 9 shows some scaling laws, demonstrating that training the model on larger datasets for longer durations and using continuous tokenization yields substantial improvements in FID.
> > For instance, in Figure 9 (Plot 3), for 32-token reconstructions, FID improves from **53.x (stage-1 on Imagenet-100) --> 30.x (stage-1 on full ImageNet-1K)**, significantly closing the gap with baseline methods (trained on full ImageNet). The second stage also results in a significant drop in FID, from **53.x (stage-1 only trained on Imagenet-100) --> 22.x (stage-1 + stage-2 trained on Imagenet-100)**. We apologize if this wasn't clear from the scaling plots.
> >
> > Another example: **VQGAN with continuous 1D tokens** results at 128 tokens (shown in Table 4, Row 4) already outperform **Titok at 128 tokens**.
> > ***
> > > Dynamic halting is unclear. Figure 1 states that dynamic halting is optional, but a clear discussion of when it should be enabled or disabled is missing. It is unclear if it is enabled for all experiments or some subset.
> >
> > \
> > We apologize for this confusion, and thank the reviewer for bringing this us.
> > For all the experiments and visualizations in the paper, **dynamic halting** was performed during the first stage of training (while learning to reconstruct the 2D tokenizer tokens, VQGAN, or VAE tokens). We labeled it as optional because it is not critical to model performance, and the network can implicitly learn to avoid attending to image tokens that are already well reconstructed. Since **dynamic halting is not a core component of the approach**, we did not perform an ablation study on it before the submission deadline. However, please find the metrics with and without dynamic halting below:
> >
> > Dynamic Halting | 32 tokens | 64 tokens | 96 tokens | 128 tokens |160 tokens | 192 tokens | 224 tokens | 256 tokens
> > |--------|---------|--------|-----------|-------|---------|--------|-----------|-------|
> > $\times$|24.38|17.00|13.49|11.53|**10.30**|9.43|8.76|8.29
> > $\checkmark$|**22.57**| **16.17**| **13.30**| 11.69| 10.22| **9.30**| **8.55**| **8.25**
> >
> > ***
> > .
> > > Complexity of the method: The proposed method is not simple, including multiple stages like learning to reconstruct pre-trained VQGAN tokens, then joint encoder-decoder training, and then GAN loss at some later stage in training.
> >
> > \
> > We believe that the lack of detailed training information in the initial submission has led to some confusion. We have also improved the architecture figure now and will share the code, which should also help understand.
> >
> > **Our training regime** first involves training only the latent-distillation modules with reconstruction-only losses, followed by incorporation of GAN loss after few epochs, when we also start fine-tuning the base image encoder / decoder.
> >
> > We agree that our current training regime for the adaptive tokenizer requires multi-stage or a curriculum as defined above. And trying just a single stage training strategy would be more simple and elegant, but unfortunately this is the case with most modern vision systems like DALLE, MUSE, MaskGiT, VQGAN, StyleGAN etc.
> > **In fact, our training schedule of turning on the GAN loss after a few epochs is inspired from VQGAN and Titok.**

---

> ### Author Response · Authors · 2024-11-21
> **Conclusion**
>
> > Minor (no effect on score):
>
> Thank you for sharing these points. We will incorporate all these changes in the final paper draft. We plan to add a small related work section for fixed-length representation learning algorithms and will cite the SPARO paper.
>
> \
> **We hope we have addressed all your queries, include the baseline requests. Please let us know if there’s anything else we can do to assist in increasing the rating.**
>
> We believe the paper has strong contributions orthogonal to Matryshoka baseline (which hasn't even been fully explored for reconstruction tasks), even having potential to be combined with our recurrent and adaptive tokenizer.**
>
> Best regards!

---

> > ### Author Response · Authors · 2024-11-23
> > **Response to Reviewer tXpo**
> >
> > We really appreciate your valuable feedback.
> > We hope we have answered all your queries. With the discussion period ending soon, please let us know if there is anything further we can do to help enhance your rating.
> >
> > Best Regards!

---

> > > ### Author Response · Authors · 2024-11-25
> > > **Response to Reviewer tXpo**
> > >
> > > We believe we have addressed all your queries –– adding training and gradient details, showcasing comparisons and benefits over the Matryoshka baseline, incorporating a dynamic halting ablation, and addressing other points raised. We will also make the minor suggested changes.
> > >
> > > As the discussion period concludes tomorrow, we would greatly appreciate your acknowledgment of our response and efforts. We are also eager to hear your feedback.
> > >
> > > Best Regards!

---

> > > > ### Comment · Reviewer_tXpo · 2024-11-26
> > > >
> > > > Thank you for all your responses. I have decided to keep my score. I'm not entirely convinced that one might want to use the paper's method over simpler, fixed-length baselines in representation learning, especially since performance often remains comparable to the baselines. Furthermore, it is unclear how the authors compare their method fairly with these baselines -- different models have different training datasets, different model sizes, and potentially different FLOPS budgets.

---

> ### Author Response · Authors · 2024-11-26
> **Response to Reviewer tXpo**
>
> Firstly, we would like to express our disappointment with the reviewer’s rating and response. At the same time, we sincerely appreciate the positive feedback from other reviewers and the broader community, who have recognized the novelty of our approach, its significance, and the comprehensive nature of our experiments.
>
> * The applications of dynamic tokenization are evident, particularly in areas such as high-resolution image and video generation, image-language, and video-language models, where fixed-length representation learning systems lead to exceedingly long context or number of tokens. Furthermore, compression (theories from Ray Solomonoff, Occam's Razor, Jurgen Schmidhuber's Low Complexity Art, Shane Legg and Marco Hutter's Universal Intelligence paper and many more) and adaptive processing are widely regarded as essential components of intelligent systems, such as OpenAI’s o1. This perspective has been reinforced by the encouraging responses we have received so far, including significant interest in scaling up the proposed approach.
>
> * Next, we firmly believe that the proposed approach is highly novel and represents an important step forward in representation learning—building adaptive representations is already becoming increasingly vital, particularly in language models. We are, however, disappointed that Reviewer tXpo does not appear to acknowledge the novelty of our approach, which has been strongly recognized and appreciated by all other reviewers.
>
> * While no existing approach applies a Matryoshka-style model to the reconstruction task (other than an ongoing ICLR submission), we implemented this baseline at the reviewer’s request. Unfortunately, it appears that the reviewer has not engaged with the results we shared and has not raised any clarifying questions to facilitate further discussion.
>
> * Finally, for clarification -- **to ensure a fair comparison, we trained both our approach and the Matryoshka-style model on the ImageNet100 dataset. We did ablations of various different sizes for both our approach and Matryoshka model as shared above. Additionally, we adjusted the Matryoshka model's size so that the training flops of the large model matches with the flops our recurrent approach.** The results shared above demonstrate that our method not only achieves comparable quantitative performance on the in-distribution validation set but also significantly outperforms the baseline on out-of-distribution datasets like COCO and Wikipedia-Text images. Moreover, **the qualitative examples provided in the updated manuscript (Figure 21) illustrate how recurrent processing yields sharper reconstructions and better object discovery—an advancement that, to our knowledge, has not been demonstrated before.**
>
> We respectfully request that this review be brought to the attention of the Area Chair, leaving it to their sound judgment. From proposing a novel approach to tokenization to supporting it with extensive experiments, **we believe we have thoroughly addressed all reviewers' questions and put forth our best efforts in this paper.**

---

> > ### Comment · Reviewer_tXpo · 2024-11-26
> >
> > Thank you for your response, which addresses my comment about the applicability of the work. Can the authors please help me with the following (more critical) concern:
> >
> > How do the authors perform a controlled and fair comparison against the baselines currently used throughout the main body of the paper, namely Titok-L-32, Titok-B-64, Titok-S-128, and VQ-GAN? What factors are kept constant across these setups (training datasets, model size, max representational capacity, FLOPS, etc.)? For those factors, why are those factors the right ones to maintain fixed for proper comparison in the context of your work?
> >
> > The discussion the authors shared about their comparison with the Matryoshka-style model in the previous comment is more convincing, but this experiment is not central to the manuscript.

---

> ### Author Response · Authors · 2024-11-26
> **Response to Reviewer tXpo**
>
> ### **Dataset**
>
> -   We trained our model on ImageNet100 (a subset of 100 ImageNet classes) and demonstrated comparable performance to models trained on the full ImageNet dataset—**a strong generalization capabilities of our approach.** As we have consistently emphasized in our responses and the main paper, as an academic institute, we lack the resources to train on the entire ImageNet dataset.
> - Secondly, as requested and acknowledged by Reviewer 4, **we trained Titok-B-64 on ImageNet100 and observed a performance drop compared to training on the full ImageNet.** This result clearly highlights the potential of scaling to larger datasets.
> - Additionally, **we presented extensive scaling laws to further support our claims.** Specifically, training even a single stage on Imagenet1K vs ImageNet100 provides significant benefits (see Fig 9, plot 3, compare circle line w/ triangle line). Second stage training (because of GAN loss) also always improves FID significantly -- *from 53.x (stage-1, IN100) -> 30.x (stage-1, full IN1K);  from 53.x (stage-1, IN100) -> 22.x (stage-1 + stage-2, IN100).*
> - In response to the reviewer's request, we also included **intriguing token-map attention comparisons with DINO in Figure 19.** For example, see the **background wall switch** discovered in the last row of the plot effectively illustrates our approach’s capabilities.
> - Beyond ImageNet100, we showcased the efficacy of our method on direct internet images (alongside OOD datasets already shown in paper like COCO, Art dataset, WIT dataset), which are substantially different from the ImageNet100 dataset, further validating the robustness of our approach.
> - Finally, **it is important to note that no adaptive tokenization baselines currently exist for the reconstruction task.** Fixed-length baselines lack the adaptability to input, a feature that is becoming increasingly desirable in large models (**big example being OpenAI o1, which performs much more desirable flops than similar other models**). While there is a slight performance drop, this is due to the amortized nature of our approach for all token-length reconstructions—an aspect we have reiterated throughout the paper and review process.
> - We firmly believe that, given the constraints of an academic setting, we have presented a strong case for the benefits of an adaptive and recurrent tokenizer. This approach is already attracting interest from industrial institutions exploring similar ideas, and we are working to scale it to larger datasets than even Imagenet1K.
>
>
> ### **Flops Matching with Fixed-Length Baseline**
>
> * We truly appreciated your suggestion to implement adaptive baselines (despite their non-existence) and your insightful recommendations on matching FLOPs. This motivated us to conduct a thorough ablation study within just a one-week period.
> * As mentioned earlier, **the goal of this paper is fundamentally different from fixed-length baselines. Our objective is to propose an adaptive tokenizer capable of assigning a variable number of tokens to each image based on its content.** This adaptive tokenizer can potentially replace existing tokenizers like VAE, VQGAN, or CLIP-Encoders used in large models such as DALL-E and Stable Diffusion, enabling greater flexibility and capability through adaptive tokens. As one of the first works in this direction, we believe that achieving comparable performance to existing 2D tokenizers while incorporating adaptiveness is a significant contribution, as acknowledged by multiple reviewers.
> * Moreover, one can leverage the best state-of-the-art base tokenizer and distill it into an adaptive tokenizer. We have already demonstrated this with both VAE and VQGAN, noting that distilling VAE even outperformed Titok, as highlighted in our earlier response.
> -   Finally, regarding your point about matching FLOPs: since our approach builds on top of a 2D tokenizer, it naturally requires slightly more FLOPs. However, **the increase compared to the baseline is minimal, as already shared in response to other reviewers' requests.** For instance, compared to Titok-L-32, our method requires just 30 GFLOPs more to output 32 tokens—a negligible difference on most modern systems capable of executing TFLOPs per second. In fact, at the 32-token level, our approach is slightly faster than Titok, which uses a 24-layer transformer, compared to our 8-layer transformer. **Each recurrent iteration takes only 4ms and an additional 40 GFLOPs, as already noted in the rebuttal.**
> -   Furthermore, as previously mentioned, there is significant potential for system-level optimizations to reduce FLOPs and accelerate each recurrent iteration by incorporating well-known techniques like KV-caching and Flash-Attention.
> * **Many recent SOTA works have started building on top of existing tokenizers to offer new capabilities. For instance, MAGE (builds on VQGAN), Titok (builds on VQGAN), MAR (builds on both VAE and VQGAN), RCG (builds on Stable Diffusion), and many more.**

---

> > ### Author Response · Authors · 2024-11-26
> > **Response to Reviewer tXpo**
> >
> > Given the original review, we made our best efforts to address all the questions you raised. **To summarize:** throughout the paper, we trained on a much smaller dataset than the fixed-length baselines and still matched their performance. Additionally, we supported all results with extensive ablations and scaling laws -- covering aspects such as model size, training duration, dataset size, and continuous vs. discrete tokens, ablations on several OOD datasets.
> >
> > We also believe that the constraints of an academic institute should not hinder valuable research. With just one H100 node shared by lot of lab members, we still managed to achieve this level of performance. With growing industry interest, we will strive to scale our approach to much larger datasets, beyond even ImageNet1K. **However, it’s important to note that scaling up is not the core research contribution of this work.**
> >
> > We hope this clarifies any doubts you have.

---

> > > ### Comment · Reviewer_tXpo · 2024-11-27
> > >
> > > > We trained our model on ImageNet100 (a subset of 100 ImageNet classes) and demonstrated comparable performance to models trained on the full ImageNet dataset—a strong generalization capabilities of our approach. As we have consistently emphasized in our responses and the main paper, as an academic institute, we lack the resources to train on the entire ImageNet dataset.
> > >
> > > It's understandable if the authors do not have the resources to train on large datasets. However, the paper's experiments should be controlled in a meaningful way to provide clear support for its message. For instance, the baselines could have a similar training setup as what ALIT is trained with, controlled to not use more FLOPS or parameters than what ALIT uses. There can also be different classes of such fixed settings, such as for smaller and larger models. When baselines get some advantages and the paper's model gets some advantages in non-systematic ways, it is hard to draw proper scientific conclusions from the results.
> > >
> > > > Additionally, we presented extensive scaling laws to further support our claims.
> > >
> > > The argument made by the authors about the baselines getting worse with smaller datasets and ALIT getting better with larger datasets requires imagining asymptotic behaviours where numbers are not readily available and make comparisons there, which can be very error-prone. These are good supporting points to address the gap between the authors' evaluation setup and the ones originally used by the baselines, but they should not be part of the core set of results supporting the paper.
> > >
> > > > Secondly, as requested and acknowledged by Reviewer 4, we trained Titok-B-64 on ImageNet100 and observed a performance drop compared to training on the full ImageNet. This result clearly highlights the potential of scaling to larger datasets.
> > >
> > > In this result, the authors show that Titok-B-64's FID changes from 8.22 to 9.82, but this value is still better than that obtained by ALIT for comparable representation sizes. It is unclear why requiring more iterations and larger representation sizes to match the baseline performance is a good thing.
> > >
> > > > We truly appreciated your suggestion to implement adaptive baselines (despite their non-existence) and your insightful recommendations on matching FLOPs. This motivated us to conduct a thorough ablation study within just a one-week period.
> > >
> > > I thank the authors for adding the Matryoshka-style baseline. This is an important experiment, and conclusions are easier to make due to it being a better controlled comparison. However, it is not the only baseline, nor is it as central as the others that raised evaluation concerns.

---

> ### Author Response · Authors · 2024-11-27
> **Response to Reviewer tXpo**
>
> > The argument made by the authors about the baselines getting worse with smaller datasets and ALIT getting better with larger datasets requires imagining asymptotic behaviours where numbers are not readily available and make comparisons there, which can be very error-prone. These are good supporting points to address the gap between the authors' evaluation setup and the ones originally used by the baselines, but they should not be part of the core set of results supporting the paper.
>
> We believe the points we have presented are robust and do not rely on speculative asymptotic behavior. The baselines in the paper were trained on a larger dataset (ImageNet-1k), whereas our model was trained on a smaller dataset (ImageNet-100). Despite this, our model achieves results that are on par with the baselines. **Our primary argument is that, when trained on the larger dataset, our model's performance will either remain at the same level in the unlikely worst-case scenario or improve (scaling laws already hint at this). In any case, the reported performance will not degrade, and all the contributions will hold.**
>
> Even if the model’s performance on the larger dataset remains consistent with the current numbers, all conclusions and contributions of our work still hold. We hope this clarification addresses any concerns.
>
>
> > In this result, the authors show that Titok-B-64's FID changes from 8.22 to 9.82, but this value is still better than that obtained by ALIT for comparable representation sizes. It is unclear why requiring more iterations and larger representation sizes to match the baseline performance is a good thing.
>
> Our primary contribution lies in adaptive tokenization, which is not a feature of Titok. As emphasized in our paper, adaptive tokenization is a highly desirable property for vision models. The reviewer notes that Titok-64 achieves a better FID than our method when our method is restricted to 64 tokens. However, **our approach is not limited to 64 tokens—the same network can seamlessly infer representations with 128 or 256 tokens, achieving an FID as low as 8.25.** In contrast, Titok would require training a separate network to achieve improved results at 256 tokens. Our model’s ability to use the same network for inference across different token lengths is a key advantage.
>
> **Why is this approach valuable?** Consider a dataset where a subset of images can be effectively represented with just 32 tokens (where “effectively” means achieving FID < some threshold), while other images require higher token lengths. Our model can dynamically allocate fewer tokens to the first subset and more tokens to the rest, optimizing both performance and efficiency. In contrast, Titok-64 is constrained to using 64 tokens for all images, which results in a higher FID compared to our model's adaptive approach. On the contrary VQGAN-256, would achieve a lower FID, but would use 256 tokens for every image, failing to adapt by using fewer tokens for simpler cases. Note: Our approach is not restricted to 256 tokens, one could train ALIT to adapt upto 1K tokens something which current large scale generative models use. **Adaptive computation is a significant and relevant challenge in the field—our method addresses it, while Titok does not.**

---

### Official Review · Reviewer_rXC5 · 2024-11-04

**Soundness:** 3
**Presentation:** 2
**Contribution:** 2
**Rating:** 6
**Confidence:** 4

**Summary:**

This paper addresses a fundamental limitation in current visual representation learning systems: their use of fixed-length representations regardless of image complexity. The authors introduce an Adaptive Visual Tokenizer (AVT) that can generate variable-length representations (ranging from 32 to 256 tokens) through a recursive processing mechanism.

The key innovation is a recurrent encoder-decoder architecture that progressively distills 2D image tokens into 1D latent tokens over multiple iterations. During each iteration, the system not only refines existing tokens but also introduces new ones, allowing for adaptive representation capacity. This approach enables tokens to specialize and focus on specific image regions, leading to emergent object discovery without explicit supervision.

The authors provide extensive empirical evidence demonstrating that optimal token counts naturally vary based on several factors: image complexity/entropy, familiarity with the training distribution, downstream task requirements, and the strength of models using these representations. Through experiments on image reconstruction, classification, and depth estimation tasks, they show that their adaptive approach achieves comparable performance to fixed-length tokenizers while offering the flexibility to use fewer tokens when appropriate.

Beyond the technical contribution, the paper makes a broader conceptual contribution by demonstrating how variable-length representations align with human visual processing and information theory principles. The work suggests promising directions for more efficient visual representation learning, particularly for applications like video understanding where fixed-length representations may be insufficient.

-----------

Update: Added response and modify score to 6, presentation to 2.

**Strengths:**

**Originality**
- Novel integration of recurrent processing with visual tokenization, extending beyond fixed architectures like VQGAN/Perceiver
- adapting dynamic computation (from NLP/sequential domains) to visual representation learning
- Fresh perspective on image compression through variable-length tokens, contrasting with traditional fixed-length approaches
- Original empirical framework for analyzing token requirements across complexity, familiarity, and model capacity

**Quality**
Thorough empirical validation across multiple dimensions:
- Reconstruction quality (L1, FID) is competitive with specialized fixed-length tokenizers
- Linear probing results comparable to larger models (49.9% Top-1 vs Titok's 48.0% with smaller architecture)
- Systematic ablations of architecture choices (continuous vs discrete, network depth, training strategies)
- Strong technical foundation combining quantized latent spaces, recurrent processing, and adaptive computation
- Rigorous analysis of token specialization through attention map visualization and segmentation alignment (57.8 mIOU without explicit supervision)

**Clarity**
- Clear architectural progression from base distillation to full adaptive system
- Well-structured empirical validation of key hypotheses about token requirements
- Effective use of visualizations to demonstrate token specialization and attention patterns
- Systematic organization of ablation studies exploring key design choices

**significance**:
- Demonstrates feasibility of variable-length visual representations while maintaining performance
- Introduces techniques potentially valuable for resource-constrained deployment scenarios

The paper's main strength lies in its systematic challenge to the fixed-length paradigm while maintaining competitive performance, backed by comprehensive empirical analysis.

**Weaknesses:**

The paper's primary weakness lies in its validation strategy and scaling limitations. While the approach shows promise, the core experiments are limited to ImageNet-100 (rather than full ImageNet-1K) for the adaptive tokenization training, making it difficult to assess the method's capabilities at scale fully. The authors acknowledge this limitation in Table 1's footnote, but don't sufficiently explore whether the performance gap with fixed tokenizers (particularly in FID scores for low token counts) is fundamental to the approach or simply due to training scale.

A second significant weakness is the lack of thorough comparison with specialized architectures. While the paper compares with fixed-length tokenizers (VQGAN, Titok), it omits comparisons with other adaptive approaches like FlexViT (variable patch sizes) or hierarchical architectures that naturally handle multi-scale features. It isn't easy to assess whether the benefits come from the variable-length representations or could be achieved through simpler adaptive mechanisms.

A third weakness is the lack of thorough investigation into computational overhead. The approach's recurrent nature requires multiple passes through the encoder-decoder architecture to generate variable-length representations. While Figure 9 includes some ablations on network depth and training, there's no detailed analysis of inference time complexity or memory requirements compared to single-pass approaches like VQGAN or Titok. This is particularly important given the paper's motivation of efficient representation learning.

Finally, while the adaptive token allocation based on complexity is well-motivated, the paper lacks a formal framework for analyzing the optimality of these allocations. The empirical correlations with human-labeled complexity scores are interesting but don't provide theoretical insights into whether the model is making optimal token allocation decisions. Given task constraints, including some theoretical analysis or bounds on optimal token allocation would significantly strengthen the paper's contributions.

**Questions:**

1. **Computational Efficiecy and scaling**
- Could you provide a detailed analysis of inference time and memory requirements compared to single-pass approaches?
- How does the recurrent processing overhead scale with a number of iterations?
- Have you explored early stopping strategies to reduce computation for "simple" images?
- What are the key bottlenecks preventing training on total ImageNet-1K?

2.**Token allocation optimality**
- Given task constraints, Is there a theoretical framework for determining the optimal number of tokens?
- How do you ensure the progressive token addition across iterations efficiently utilizes capacity?
- Have you explored methods to predict the required token count before processing, similar to adaptive computation time in transformers?
- Could you provide an empirical analysis showing token utilization across iterations (e.g., are later tokens less "important")?

3. **Comparison with specialized architectures**:
- How does the method compare with recent adaptive patch-size approaches (like FlexViT) on standard benchmarks?
- Could you provide direct comparisons with self-supervised methods like DINO on standard metrics for object discovery capabilities?
- Have you explored combining your approach with hierarchical architectures that naturally handle multi-scale features?

4.  **Codebook and quantization design**
- What motivated the choice of shared codebook across iterations versus hierarchical/specialized codebooks?
- How sensitive is the method to codebook size and quantization strategy?
- Could you analyze codebook utilization across different iterations and image complexities?
- Have you explored using learned or adaptive quantization strategies?

These clarifications would help comprehensively assess the method's practical applicability and theoretical foundations.

---

> ### Author Response · Authors · 2024-11-20
> **Address all questions regarding **computational efficiency and scaling:****
>
> We sincerely thank the reviewer for acknowledging our paper with such a detailed strength section. We now try to answer all your queries:
>
> > Could you provide a detailed analysis of inference time and memory requirements compared to single-pass approaches? How does the recurrent processing overhead scale with a number of iterations?
>
> \
> Thanks for the great question!\
> The overhead of running an additional iteration of the adaptive tokenizer encoder is actually quite small: **4ms run time and <+30 gflops on a single h100 gpu**. Here are the comparisons with a fixed length baseline:
>
> \
> Inference time comparison (in milli-seconds, on single h100 gpu, fp32 precision) for single image encoding:
> Algorithm | Iteration 1 | Iteration 2 | Iteration 3 | Iteration 4 | Iteration 5 | Iteration 6 | Iteration 7 | Iteration 8
> |----------|----------|----------|----------|----------|----------|----------|----------|----------|
> Titok-l-32| 9.54
> Ours-S| 6.84|10.40|14.13|18.04|22.03|25.99|30.20|34.74
>
>
> \
> GFlops Comparison (on h100 gpu) for single image encoding:
> Algorithm | Iteration 1 | Iteration 2 | Iteration 3 | Iteration 4 | Iteration 5 | Iteration 6 | Iteration 7 | Iteration 8
> |----------|----------|----------|----------|----------|----------|----------|----------|----------|
> Titok-l-32| 58.44
> Ours-S| 83.88|105.45|129.16|155.03|183.04|213.21|245.53|278.00
>
> \
> **Some points regarding the tables:**
> * For 32 tokens, we our latent-distillation encoder has 8 transformer layers much smaller than Titok-L-32 (with same number of attention heads, feature dim and same number of tokens). Despite that, we have more GFlops because of using  VQGAN image encoder + a small additional patch embedding layer as the image 2D embedding. We can switch to Titok-L-32 style image encoding and reduce upto 50 gflops.
> * Overall, +30 gflops overhead per iteration is very less for modern compute and can be further reduced by using techniques like KV caching previous layer attentions etc.
> * The reported metrics for inference time are using fp32 precision. With lower precision, inference time can be further lowered without loss in accuracy.
>
> We are working towards scaling the adaptive tokenizer and making it more efficient from systems perspective. Hope this answers your question.
> ***
>
> > Have you explored early stopping strategies to reduce computation for "simple" images?
>
> \
> Thank you for the great suggestion!
>
> **Automatic token selection per-image at Inference**: We already presented an automatic token selection criterion in the paper (via Figure 5) but, on reflection, didn’t explicitly highlight it as “automatic.” The criterion is based on a reconstruction loss threshold: we decode the image using varying token counts and select the minimum token length that satisfies the objective of reconstruction loss < threshold. This approach ensures an efficient allocation of tokens while meeting quality requirements. Additionally, we will provide this automatic token selection as a user-friendly API in the released code.
> **Figures 3, 10, and 11 demonstrate that reconstruction loss correlates strongly with image entropy**, making token selection by reconstruction loss effectively equivalent to selection based on image entropy. Furthermore, we tested our adaptive tokenizer on internet images (Figure 20)—significantly different from the ImageNet100 dataset used for training—and observed that the automatically selected (variable) token counts effectively represent these diverse input images.
>
> **Early stopping during training** is another great suggestion. Exploring different training curriculums, such as traditional hard-negative mining, to enable early stopping would indeed be an intriguing direction for future work. Currently, during training, we sample all images in the dataset equally.
>
>
> ***
> > What are the key bottlenecks preventing training on total ImageNet-1K?
>
> \
> The main bottleneck preventing training on the full ImageNet-1K dataset is compute constraints. Our access is primarily limited to a single H100 machine shared among multiple students. Training on the full ImageNet dataset would require exclusive use of the machine for several days, a challenge faced by many current vision systems.
> Our primary focus has been on proposing a novel recurrent and adaptive tokenizer and investigating the architectural insights of this model. We truly appreciate that you highlighted this in the detailed strengths section of your review.
> That said, we have demonstrated throughout the paper that our ImageNet100-trained model generalizes remarkably well to various out-of-distribution (OOD) images. **Additionally, we have already presented scaling laws by running one of the training stages on the full ImageNet dataset.** Lastly, we are grateful for the widespread interest we received in scaling the proposed tokenizer to much larger datasets, and are working on it.

---

> ### Author Response · Authors · 2024-11-20
> **Addressing all questions regarding token allocation**
>
> > Given task constraints, Is there a theoretical framework for determining the optimal number of tokens?
>
> We thank the reviewer for the great question! Deriving a theoretical bound for the optimal number of tokens for an image would be an intriguing avenue for future research but is beyond the scope of this work. More broadly, determining the optimal compression factor for an image has remained an unsolved challenge for decades, highlighting its complexity as a theoretical problem.
> ***
> > How do you ensure the progressive token addition across iterations efficiently utilizes capacity?
>
> The image reconstruction figures and metrics (Figures 2, 3, 9; Tables 1), along with downstream task performance plots (Figure 6) strongly demonstrate how progressively adding new tokens effectively leverages the model's capacity. Iteratively improving current tokens while adding new ones leads to sharp and specialized attention maps as in Figure 8, 15.
> ***
> > Have you explored methods to predict the required token count before processing, similar to adaptive computation time in transformers?
>
> To the best of our knowledge, no learned method currently exists that can predict the required token count for each image in a generalizable manner. That said, one could **train a small neural network on the pseudo-ground truth data generated by our adaptive tokenizer**. The pseudo-ground truth data could take the form: (image, token count, reconstruction loss predicted by the adaptive tokenizer).
> ***
> > Could you provide an empirical analysis showing token utilization across iterations (e.g., are later tokens less "important")?
>
> Thank you for the thoughtful question! The reconstruction L1 loss and FID metric at different token counts provide an empirical analysis of token utilization across iterations—demonstrating that FID improves progressively as the token count increases (e.g., from 32 to 64 to 128, and so on).
>
> Visually (as shown in Figure 12, 15 via attention map visualizations) and intuitively, when the representational capacity is limited (e.g., at 32 tokens), the first 32 tokens must cumulatively attend to all image regions to ensure accurate reconstruction. As additional tokens are introduced over iterations, the first 32 tokens gradually specialize, attending to specific regions, which improves unsupervised segmentation (Table 2). Meanwhile, the latter tokens contribute additional representational capacity, focusing on regions requiring more attention.
>
> **Figure 12 illustrates this process**, showing how the attention maps of the first 32 tokens become increasingly sparse and precise over iterations, from 1 to 8. We hope this explanation clarifies how token utilization remains high and effective across multiple iterations.

---

> ### Author Response · Authors · 2024-11-20
> **Addressing all questions regarding comparison with specialized baselines**
>
> >   How does the method compare with recent adaptive patch-size approaches (like FlexViT) on standard benchmarks?
>
> \
> We sincerely thank the reviewer again for this thoughtful suggestion!
>
> FlexVIT was originally designed for classification tasks, and to the best of our knowledge, **no research has explored adapting it to reconstruction or adaptive generative tasks**, making it a non-trivial open research challenge. Nonetheless, as requested by the reviewer, **we developed a FlexVIT-based reconstruction baseline using our adaptive tokenizer style decoding (using our proposed latent distillation decoder)** and train /  evaluate two versions—FlexVIT-Small and FlexVIT-Large—on the ImageNet100 dataset for a fair comparison. Please find the FID metrics at different token counts below:
>
> Algorithm | 32 tokens | 64 tokens | 96 tokens| 128 tokens | 160 tokens| 192 tokens|224 tokens| 256 tokens
> |--------|---------|--------|-----------|-------|--------|-----------|-------|-------|
> FlexVIT-S| 79.84|39.73|-|15.16|-|-|-|12.64
> Ours-S|**22.57**|**16.17**|**13.30**|**11.69**|**10.22**| **9.30**| **8.55**|**8.25**
> FlexVIT-L| 23.30|14.73|-|**9.92**|-|-|-|**7.80**
> Ours-L|**19.70**| **13.92**|**11.39**|10.41|**9.23**|**8.75**| **8.22**|8.03
>
> Some points regarding FlexVIT:
> * Firstly, this version of FlexVIT is using our adaptive tokenizer style image decoding on top of FlexVIT Encoder.
> * We perform better than FlexVIT at lower token counts and on-par with larger token counts (delta < 0.5 FID).
> * Since FlexVIT samples tokens by varying patch size, encoding the image into arbitrary token count, which is not a factor of original image size / token count, is not possible.
> * The **"adaptive" contribution of FlexVIT (adaptive in terms of input patch size) is orthogonal to our work**. Our adaptive tokenizer could indeed be extended to variable-resolution images by incorporating FlexVIT’s variable patch-size mechanism.
> * Mapping an image to a variable-sized representation using FlexVIT requires running it multiple times with different patch sizes, with no computation or FLOPs shared between runs. In contrast, our proposed recurrent tokenizer does computation sharing between different token executions, thus providing more **efficient FLOPs per image** to determine the optimal token count.
> * FlexVIT encoder only has patch-bound tokens and no global tokens, so **token visualization or unsupervised object discovery is not straight-forward** (there is also no class token like DiNO).
>
> ***
> >   Could you provide direct comparisons with self-supervised methods like DINO on standard metrics for object discovery capabilities?
>
> \
> As acknowledged by the reviewer, we believe our contributions are independent of those of DINO. DINO is a fixed-length representation learning algorithm, with its key contribution being the self-distillation self-supervised learning technique. As demonstrated in the DINO paper, token-attention maps are generated by attending to the class token’s attention heads for tokens linked to different 2D patches of the image. Since DINO-S has only six attention heads, this results in six attention maps. Moreover, because DINO’s objective is implicitly tailored for classification, the attention maps prioritize regions that enhance classification performance. **As suggested, we compare our learned token-attention maps with the corresponding maps generated by DINO for object discovery capabilities in Figure 19.**
> ***
> >   Have you explored combining your approach with hierarchical architectures that naturally handle multi-scale features?
>
> \
> Thanks for another great suggestion! We believe that our contribution is orthogonal to general vision models with hierarchical architectures. We are multi-level in terms of number of tokens, the goal of each iteration is to best represent the input image, so at test time different token counts could be selected for different images. It would be interesting future research to combine hierarchal feature learning with the adaptive tokenizer.

---

> ### Author Response · Authors · 2024-11-21
> **Addressing questions regarding codebook and quantization design & Conclusion**
>
> > What motivated the choice of shared codebook across iterations versus hierarchical/specialized codebooks?
>
> \
> We ablate the use of shared vs different or specialized codebooks at each iteration in the following table (using reconstruction FID metric).
>
> Codebook Size | Codebook Sharing|32 tokens | 64 tokens | 96 tokens | 128 tokens |160 tokens | 192 tokens | 224 | 256 tokens
> |--------|---------|--------|-----------|-------|---------|--------|-----------|-------|-------|
> 4096| $\times$| 26.84|18.71|14.71|12.82|10.91|10.02|9.05|8.61
> 4096|$\checkmark$|**22.57**| **16.17**| **13.30**| **11.69**| **10.22**| **9.30**| **8.55**| **8.25**
>
> Some points:
> * The above table shows the benefit of shared codebook.
> * Furthermore, shared codebook leads to (a) more compression in terms of total codes to be saved. (b) theoretically allows one to rollout the adaptive tokenizer for much longer rollouts than done at train time. This is something we are concurrently looking into, and will update findings in final paper. (Another strong advantage of the proposed recurrent tokenizer)
> ***
> >   How sensitive is the method to codebook size and quantization strategy?
>
> \
> We ablate the performance based on codebook size below (using reconstruction FID):
>
> Codebook Size | 32 tokens | 64 tokens | 96 tokens | 128 tokens |160 tokens | 192 tokens | 224 tokens | 256 tokens
> |--------|---------|--------|-----------|-------|---------|--------|-----------|-------|
> 2048| 26.65|18.91|14.76|12.20|10.86|10.03|9.49|9.13
> 4096|**22.57**| **16.17**| 13.30| 11.69| 10.22| 9.30| **8.55**| **8.25**
> 8196|23.76| 16.55| **13.11**| **11.14**| **9.75**| **9.17**| 8.61| 8.42
>
> Some points:
> * As can be seen, there is not much (marginal) difference  in performance between 4096 and 8196 codebook size, stating that the tokenizer is less sensitive to the codebook size. That said, when scaling to large models, scaling up the codebook while maintaining utilization would be an intruiging and open research question. 4096 is a usual codebook size for Imagenet models.
> * Secondly, just to highlight we have also shown experiments with no 1D quantization at all in the supplementary and the main paper Figure 9.
> ***
> >   Could you analyze codebook utilization across different iterations and image complexities?
>
> \
> This is a great question! Through the new Figure 22 in the updated paper, we showcase codebook utilization over the 5000 images of the Imagenet100 validation set. We are quite happy with the visualization (and thank the reviewer for the suggestion) -- as it clearly highlights that at larger rollout iterations, some of the codes are sampled much more regularly, highlighting code / token specialization. Thus backing our core intuition -- **When you have more memory / tokens, some of the memory can be utilized for specialization.**
> ***
> >   Have you explored using learned or adaptive quantization strategies?
>
> \
> We explored several quantization techniques, including: (a) **factorization**, where a larger embedding is mapped to a smaller embedding before searching the codebook for the closest neighbor; (b) first training the encoder-decoder architecture with continuous embeddings for one or more epochs, and then switching to quantized embeddings **after initializing the codebook via k-means sampling of continuous codes**; and (c) performing **non-gradient-based EMA updates of the codebook** instead of using gradient-based optimization with the straight-through estimator.
>
> Ultimately, we found that factorization combined with gradient-based updates of the codebook (using uniform initialization of the codebook codes) worked best, so we chose to stick with this approach. Testing other quantization techniques could be an interesting direction for future research, and we thank the reviewer for this suggestion. All said, we already provided detailed experiments comparing continuous vs discrete tokenization at both 2D and 1D levels through Figure 9 plot 4 and Figure 18.
>
> \
> **We hope we have addressed all your queries. Please let us know if there's anything else we can do to assist in increasing the rating, more in-alignment with the detailed strengths section you provided.**
>
> Best regards!

---

> > ### Author Response · Authors · 2024-11-22
> > **Response to Reviewer rXC5**
> >
> > We really appreciate your strong and detailed positive feedback, as well as your valuable suggestions for ablations.
> > We hope we have answered all your queries. With the discussion period ending soon, please let us know if there is anything further we can do to help refine your rating, more in alignment with the strengths you mentioned.
> >
> > Best Regards!

---

> ### Author Response · Authors · 2024-11-24
> **Theoretical Formulation for Optimal Token Bound**
>
> > Given task constraints, including some theoretical analysis or bounds on optimal token allocation would significantly strengthen the paper's contributions.
>
> \
> Our adaptive length image tokenizer is akin to an auto-encoder, where we distill the original 2D image tokens (say, $X$) into 1D latent tokens (say, $Z$) iteratively, with the goal of reconstructing $X$ from $Z$. In each iteration, we increase the dimensionality of $Z$ in terms of the number of tokens used to represent $Z$, with earlier iterations mapping $X$ into the most compressed $Z$. Thus, the theoretical formulation of our adaptive tokenizer can be written down using the seminal rate-distortion theory:
>
>
> #### **Defining Rate-Distortion Terms:**
>
> - Rate $R(Z)$: The number of tokens for the representation ($Z$) used to represent $X$ (image or 2D image tokens). Mapping from X to Z is performed by the **latent-distillation encoder**.
> - Distortion $D(X, \hat{X}(Z))$: The reconstruction loss between $X$ and $\hat{X}(Z)$, i.e., the reconstruction of $X$ from $Z$ performed by the **latent distillation decoder**.
>
> The distortion is computed as:
>
> $$
> D(X, \hat{X}(Z)) = \frac{1}{N} \sum_{i=1}^{N} \left( X_i - \hat{X}_i(Z) \right)^2
> $$
>
> Rate $R(Z)$ can be expressed as the mutual information:
>
> $$
> R(Z) = I(X; Z) = H(X) - H(X | Z)
> $$
>
> #### **Objective Function:**
>
> We aim to minimize the rate-distortion trade-off, represented as:
>
> $$
> \min_{Z} \left[ R(Z) + \lambda D(X, \hat{X}(Z)) \right]
> $$
>
> This can also be expressed as:
>
> $$
> \min_{Z} \left[ I(X; Z) + \lambda D(X, \hat{X}(Z)) \right]
> $$
>
> $$
> \min_{Z} \left[ H(X) - H(X | Z) + \lambda D(X, \hat{X}(Z)) \right]
> $$
>
> $$
> \min_{Z} \left[ - H(X | Z) + \lambda D(X, \hat{X}(Z)) \right]
> $$
>
> Thus, theoretically, our training objective minimizes the number of tokens required to represent $Z$, while keeping the distortion under control.
>
> #### **Optimal Token Bound:**
>
> The optimal tokens $Z_{\text{optimal}}$ can be found under a given distortion threshold – the optimal number of tokens corresponds to the minimal rate $R(Z)$ that satisfies the distortion constraint $D(X, \hat{X}(Z)) \leq \epsilon$, where $\epsilon$ is the acceptable reconstruction loss. This leads to the optimal token allocation as:
>
> $$
> Z_{\text{optimal}} = \min_{Z} \left[ - H(X | Z) + \lambda D(X, \hat{X}(Z)) \right]
> $$
>
> - $H(X | Z)$ is the conditional entropy, representing how much uncertainty remains in $X$ after observing $Z$.
> - The reconstruction loss $D(X, \hat{X}(Z))$ is controlled by the parameter $\lambda$, ensuring a trade-off between the number of tokens and the quality of reconstruction.
>
> #### **Conclusion:**
>
> - By minimizing the objective function, we derive the **optimal token allocation** $Z_{\text{optimal}}$ for a given image $X$, balancing the rate (number of tokens) and the distortion (reconstruction loss).
> - Our **Automatic Token Selection Criteria** as explained in the previous response performs this task by sampling different number of tokens for the latent representation Z and then selecting the most compressed representation which satisfies the reconstruction loss < threshold.
>
> Further theoretical analysis is beyond the scope of the paper, but would be an intriguing future direction.
> We thank the reviewer for this great suggestion!
>
> Best Regards!

---

> ### Comment · Reviewer_rXC5 · 2024-12-03
> **Updated score based on rebuttal**
>
> Thank you for providing detailed responses to the feedback. After reviewing the responses, I found that a few of my concerns and questions have been addressed, and I'm updating my score of 6 to reflect that. I encourage the authors to consolidate feedback from other reviewers, particularly around adding appropriate baselines and streamlining the results to make the core contributions more solid.

---

### Author Response · Authors · 2024-11-20
**General Response -- Acknowledging Reviewer's Positive Feedback**

**Overview of the response**: We first summarize the major contributions as acknowledged by the reviewers, following that we share the major edits done to address all the reviewer queries. We then answer each reviewer individually.

We sincerely thank all the reviewers for their valuable feedback and are delighted with the positive response. Summarizing reviewer feedback, using their own words --

* **Novelty**: A new approach to variable-length visual token representation (rXC5, 46Y3, 6BgE) with the novel integration of recurrent processing (rXC5, 46Y3). Fresh perspective on image compression (rXC5), offering an original empirical framework for analyzing token requirements (rXC5). Strong technical foundation combining quantized latent spaces, recurrent processing, and adaptive computation (rXC5).
* **Significance**: Variable-length visual representations for resource-constrained deployment (rXC5). Highlights the emergence of semantically meaningful tokens, enabling object discovery (rXC5, 46Y3, tXpo). Opens new avenues for image understanding, representation learning and various computer vision tasks (rXC5, 46Y3).
*   **Presentation (Writing)**: Clear and concise writing with effective use of visualizations and ablations exploring key design choices (rXC5, 46Y3, 6BgE).
* **Presentation (Experimentation)**: Thorough analysis of the adaptive representation capacity with reconstruction quality (L1, FID) competitive with specialized fixed-length tokenizers and linear probing results comparable to larger models (rXC5, 46Y3). Unsupervised high mean IOU achieved in attention map alignment with ImageNet-S GT segmentation (rXC5, 46Y3, tXpo).

---

> ### Author Response · Authors · 2024-11-20
> **General Response – Major updates to the paper, answering all the reviewer questions**
>
> We now report the major changes done to the paper addressing **all the reviewer's thoughtful suggestions and queries**:
>
> * **Updated Title** -- Adaptive Length Image Tokenization via Recurrent Allocation
> * **New Figure 16** as overview of the proposed recurrent and adaptive tokenizer. We will make this the teaser Figure 1 in final version.
> * Improved and simplified architecture figure, **Figure 1**.
> * Training Details in Appendix A.3.
> * **New Figure 17** in Appendix showcasing reconstruction quality improvement on scaling model size.
> * **New Figure 18** in Appendix comparing continuous vs quantized tokenization.
> * **New Figure 19** in Appendix comparing DINO and Our Tokenizer token attention maps.
> * **New Figure 20** in Appendix showcasing results on random internet images (using model trained only on Imagenet100).
> * **New Baselines** -- Implemented prior adaptive tokenization / representation works -- FlexVIT and Matryoshka model, which were originally proposed for classification / retrieval, for reconstruction task using the proposed adaptive tokenizer style decoding.  Comparison against these baselines on Reconstruction FID metrics.
> * **New Figure 21** comparing baseline's token attention maps with our recurrent tokenizer attention maps. Thanks to recurrence, our adaptive tokenizer leads to sharper attentions.
> * **New Figure 22** studying codebook utilization over multiple recurrent iterations of the tokenizer.
> * **Writing Fixes** (as suggested by reviewers): Clarifying the approach for automatic token selection per image using reconstruction loss as self-supervised metric.
> * **Additional ablations** for quantization codebooks, dynamic halting etc as suggested by reviewers.
> * **Results on Imagenet1K** -- https://openreview.net/forum?id=mb2ryuZ3wz&noteId=qyPXkVBeJo (required extensive compute, added on Dec 2nd)

---

### Author Response · Authors · 2024-12-02
**General Response - Imagenet1K Results**

We would like to thank the reviewers for their positive overall remarks and efforts in reviewing our paper. We are also pleased with the general response received and are excited to build upon the proposed adaptive tokenizer.

Throughout the paper and our rebuttal, **we presented several compelling and robust results:** generalization to OOD/Internet images, key factors for image's representation capacity (paper Sec. 4), intriguing object discovery (including comparisons to DINO), scaling laws for larger datasets / extended training durations, and a comprehensive ablation study of the entire pipeline. https://openreview.net/forum?id=mb2ryuZ3wz&noteId=5XPec2NSTz

**As shown in our paper, we performed slightly better than fixed-length baselines on the Reconstruction Loss metric (Figure 2). On the FID metric, we demonstrated comparable performance to fixed-length baselines.** Our adaptive tokenizer's latent-distillation modules were trained solely on the ImageNet100 dataset due to compute constraints in an academic setting. We believe that the scaling laws presented in Figure 9 demonstrated that increasing the dataset size (among other techniques) will further reduce the FID gap or, at worst, maintain the current performance.

As our final effort, we invested on larger compute resources to **one-shot train ALIT on the full ImageNet1K dataset, without any hyperparameter tuning (e.g., codebook size, learning rate, batch size, minimum token count per image etc.), which could be critical for performance improvement in general. All settings are same as the baselines.**

We first compare against fixed-length baselines on the Reconstruction Loss metric. **Note:** The desired output of the tokenizer is to preserve as much information about the image as possible (aligned with the rate-distortion principle), rather than ensuring photo-realism if the content changes. On this note, reconstruction loss may serve as a stronger signal than the FID metric.


 **Reconstruction Loss Metric (L1 Loss * 100) Comparison**

Algorithm | Dataset | 32 tokens | 64 tokens | 96 tokens | 128 tokens |160 tokens | 192 tokens | 224 tokens | 256 tokens
|--------|---------|---------|--------|-----------|-------|---------|--------|-----------|-------|
Titok-L-32|Imagenet1K|12.29
Titok-B-64|Imagenet1K||**9.72**
Titok-S-128|Imagenet1K|||9.72
VQGAN|Imagenet1K||||||||7.99
Ours-S|Imagenet100|11.77|10.13|9.26|8.77|8.32|7.95|7.81|7.62
Ours-SemiLarge|Imagenet100|11.81|10.09|9.16|8.62|8.36|8.01|7.79|7.63
Ours-S|Imagenet1K|**11.50**|9.81|**8.78**|**8.36**|**8.01**|**7.65**|**7.53**|**7.48**

**FID Comparison**

Algorithm | Dataset | 32 tokens | 64 tokens | 96 tokens | 128 tokens |160 tokens | 192 tokens | 224 tokens | 256 tokens
|--------|---------|---------|--------|-----------|-------|---------|--------|-----------|-------|
Titok-L-32|Imagenet1K|**11.60**
Titok-B-64|Imagenet1K||**8.22**
Titok-S-128|Imagenet1K|||**8.22**
VQGAN|Imagenet1K||||||||**7.04**
Ours-S|Imagenet100|22.57|16.17|13.30|11.69|10.22|9.30|8.55|8.25
Ours-S|Imagenet1K|22.69|14.99|11.97|10.17|9.54|8.85|8.48|8.02
Ours-SemiLarge|Imagenet100|19.70|13.92|11.39|10.41|9.23|8.75|8.22|8.03

**Note:** Small (S) = 8 layers, Base (B) = 12 layers, SemiLarge = 16 layers, Large (L) = 24 layers

As shown in both tables, **our approach performs comparably to fixed-length baselines on both reconstruction loss and the FID metric, while offering additional capability of adaptive tokenization**.  For reconstruction loss, our performance is slightly better, while for FID, we are marginally behind. Throughout the paper and rebuttal, we have already highlighted several ways in which performance can be further improved.

Finally, we would like to emphasize that, just as diffusion models did not initially surpass GANs in their first iterations, our work is among the first in adaptive tokenization. Despite this, we achieve performance comparable to fixed-length baselines.

We hope this clarifies any remaining questions.

Best Regards!

---

### Meta-Review · Area_Chair_tpoH · 2024-12-20

**Metareview:**

This paper tackles the very interesting problem of allocating the right amount of bandwidth in transformer models for images. In most applications, computer vision problems are tackled with fixed sequence length models, tiling the image into a regular grid of patches. In many cases this scenario is suboptimal, as in some cases processing would benefint more, or less tokens in the sequence. The paper introduces a recurrent encoder-decoder architecture that recursively encodes the 2d images into a sequence of 1d tokens. The proposed architecture is evaluated for pixel-level reconstruction training - something like an autoencoder. The reviewers raised many points about the empirical validation, and the authors have provided ample answers addressing most of the reviewer's doubts. While the scale of the empirical validation could further be improved, I recommend this paper for acceptance.

**Additional Comments On Reviewer Discussion:**

The authors provided very ample and detailed answers to questions, which had a positive effect on ratings. One 3 became a 5, and one 5 became a 6. All reviewers acknoweldged the quality of answers in the rebuttal.

---

### Decision · Program_Chairs · 2025-01-22

Accept (Poster)